# Re:Form — Reducing Human Annotations in Scalable Formal Software Verification with RL in LLMs: A Preliminary Study on Dafny

**Chuanhao Yan**[1,3,†,*]   **Fengdi Che**[2,△,*]   **Xuhan Huang**[1,4,†,*]   **Xu Xu**[1,5,†,*]
**Xin Li**[1,6,†,*]   **Yizhi Li**[7,†,*]   **Xingwei Qu**[7,†,△,*]   **Jingzhe Shi**[1,3,†]
**Chenghua Lin**[7]   **Yaodong Yang**[9]   **Binhang Yuan**[5]   **Hang Zhao**[3,1]
**Yu Qiao**[1]   **Bowen Zhou**[1]   **Jie Fu**[1,△,◇]

[1]**Shanghai AI Lab**                                    [2]**University of Alberta**
[3]**Tsinghua University**                               [4]**Chinese University of Hong Kong, Shenzhen**
[5]**Hong Kong University of Science and Technology**   [6]**Nanyang Technological University**
[7]**University of Manchester**                          [9]**Peking University**
[*]**Equal contribution**   [†]**Work done during internship at Shanghai AI Lab**
[△]**Tech Lead**   [◇]**Corresponding Author**

**Reviewed on OpenReview:** `https://openreview.net/forum?id=cAQmIS4GOe`

## Abstract

Existing informal language-based (e.g., human language) Large Language Models (LLMs) trained with Reinforcement Learning (RL) face a significant challenge: their verification processes, which provide crucial training signals, are neither reliable nor scalable. In fact, the prevalent large proprietary models could hardly generate verifiable programs. A promising yet largely uncharted alternative is formal language-based reasoning. Grounding LLMs in rigorous formal systems where generative models operate in formal language spaces (e.g., Dafny) enables the automatic and mathematically provable verification of their reasoning processes and outcomes. This capability is pivotal for achieving large-scale, reliable formal software verification. It is a common practice to employ human-annotated chain-of-thought and answers to induce the reasoning and coding capabilities of LLMs. Unfortunately, it becomes unacceptably all-consuming to provide such priors for supervising complex programming tasks. In this work, we systematically explore ways to reduce human annotations with the formal language, Dafny, as the main environment for our pilot study. Our pipeline mainly relies on introducing an automatic and scalable data curation pipeline, and careful RL designs integrated with feedback from the formal language verifier. We introduce DafnyComp, a benchmark of compositional formal programs with auto-formalized specifications for specification reasoning. Our supervised fine-tuning (SFT) stage enables even small models (e.g., 0.5B) to generate syntactically valid and verifiable Dafny code, surpassing proprietary models. RL with regularization further improves performance, achieving stronger generalization to out-of-domain tasks and outperforming all strong baselines on the challenging DafnyComp benchmark. Anonymized code and models are available at `https://github.com/Veri-Code/ReForm` and `https://huggingface.co/Veri-Code`.

## 1   Introduction

Coding agents draw attention in the AI community amid claims that their emergent problem-solving abilities may foreshadow broader general intelligence, since coding allows interaction with the real world (Silver & Sutton, 2025), enforces deductive formal reasoning (Szegedy, 2020; Li et al., 2025a), and gives the ability of compositionality to extreme generalization (Chollet, 2019; Li et al., 2024; Tang et al., 2024). Despite the impressive progress in automated code generation due to recent advances in large language models

(LLMs) (AlphaCode Team, 2023; Li et al., 2022; Svyatkovskiy et al., 2020), ensuring the correctness of such code remains a significant challenge (Dalrymple et al., 2024) — especially in safety-critical domains such as healthcare, finance, and autonomous systems, where silent failures can have serious consequences. Traditional safeguards such as unit testing or manual code review are inherently limited: they may miss edge cases, fail to cover all execution paths, or rely heavily on human expertise. Instead, formal verification offers a principled alternative. Misu et al. (2024) suggest expressing a program's intended behavior as formal specifications and verifying whether the code can be proved correct against the formal specifications. But this alone can be insufficient: code proven against a specification may still exhibit uncaptured behaviors outside the specification's stated input domain. Therefore, we propose to independently auto-formalize the natural language query and the code, and then verify their derived specifications' equivalence, to guarantee behavioral alignment (Sun et al., 2024). This report targets a challenging subproblem: the formal specification generation, requiring deep semantic understanding and exhaustive behavioral description of arbitrary code.

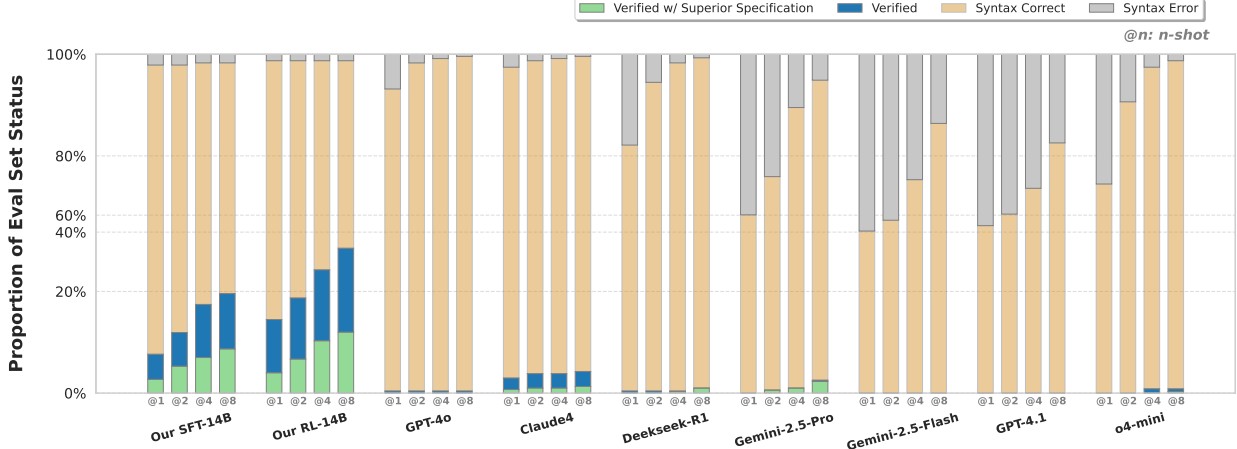

Figure 1: Model Performance on DafnyComp benchmark. We roughly categorize the cases into four groups in an ascending quality order: Syntax Error, Syntax Correct, Verified and Verified with Superior Specification. The growing proportion of Verified with Superior Specification suggests a rudimentary of exploration capabitliy to generate stronger specifcation than the ground-truth of the models incentivized by reinforcement learning. For further model behavioural analysis, a more refined set of definition of benchmarking metrics is provided in Section 3.1 as standard protocol.

A key question emerges: how can formal verification be achieved more systematically through computational approaches, potentially discovering verification strategies that complement human expertise? Unlocking this potential of scalable computational approaches (Sutton, 2019) remains difficult, primarily due to the extreme data scarcity (Thakur et al., 2025; Dougherty & Mehta, 2025). This scarcity causes even powerful LLM models, including GPT (Achiam et al., 2023), Gemini (Gemini Team, Google, 2025), Deepseek (Guo et al., 2025) and Claude (Anthropic, 2025), to perform poorly on our task as revealed in Figure 1, necessitating the development of a specific data curation and training pipeline. Looking at prevailing practice, training heavily relies on extensive and costly human annotations: models are anthropomorphized to mimic human thought processes (Ibrahim & Cheng, 2025) and finetuned to match human preference (Ouyang et al., 2022). Such reliance may trap an agent in a "cocoon" without showing genuine reasoning (Shojaee et al., 2025; Varela et al., 2025) and deriving its own strategy (Mancoridis et al., 2025). Furthermore, we cannot expect to scale up the human annotation process easily. For example, annotating formal code specifications for 50 entry-level programs can take two computer scientists approximately 220 hours (Misu et al., 2024; Austin et al., 2021), while the cost of proving SeL4 (Klein et al., 2009) is about 20 person-years. Considering these difficulties, Silver & Sutton (2025) propose a shift from human data-centric to a more scalable paradigm where learning agents get trained on their own experience (Silver et al., 2021).

Therefore, our report aims at minimizing human priors[1] and relies on reinforcement learning (RL) for open-ended exploration, uncovering novel solutions without direct human supervision. The verification-aware language Dafny[2] is an ideal environment for our pilot study because its automated verifier provides a machine-checkable correctness signal for reinforcement learning, directly addressing the difficulty of authoring formal proofs and specifications beyond human knowledge (Novikov et al., 2025). First, we **automatically generate formal specifications using proprietary frontier LLMs** to seed our training data, anticipating RL to progressively improve solution quality. To further reduce reliance on human labour, we build a pipeline to synthesize formal code by assembling current programs. The resulting synthetic dataset is held for out-of-domain generalization testing. Next, lacking a clear template for the intermediate reasoning steps needed in formal verification, we have chosen to **eliminate natural-language chain-of-thought (CoT)** from our pipeline, supported by evidence that no chain-of-thought mode suffices for certain reasoning tasks (Ma et al., 2025). Note that our goal is not to show that eliminating CoTs outperforms using CoTs, but rather to support our pipeline with minimal human annotation. Furthermore, using natural language CoT for coding with LLMs is analogous to natural language programming, which Edsger W. Dijkstra critically examines in (Dijkstra, 1979), highlighting some potential challenges related to ambiguity and precision. Finally, **our RL feedback comes from world signals or system proxies** (Silver et al., 2021; Schaul, 2024): by operating entirely in a formal-language space, an automatic evaluation signal naturally emerges (Yang et al., 2024; Misu et al., 2024), which is the correctness of formal statements. Moreover, inspired by the recent success of Goedel-Prover-V2 (Lin et al., 2025), which achieves performance comparable to DeepSeek-Prover-671B (Ren et al., 2025) on MiniF2F (Zheng et al., 2022) using only an 8B model, we believe that small models are sufficient for reasoning tasks within specific domains, such as code and mathematics. Therefore, we focus our training efforts on smaller models, ranging from 0.5B to 14B in size.

While our goal is to reduce human priors, we recognize that an entirely self-contained system without human data would be infeasible. Without any inductive bias, an RL agent starts by treating all token sequences equally, causing the subsequent exploration to be highly sample-inefficient (Mitchell, 1980). In practice, the foundational biases encoded in LLMs have driven their breakthroughs in informal reasoning tasks (Petty et al., 2025; Ruis et al., 2025). Accordingly, we retain the following human priors while aiming to minimize reliance on human annotations:

- training data seeding at the existing Python code for generating formal specifications,
- a base model pre-trained on massive human data,
- a limited supervised fine-tuning process, and
- human-designed reward, but based on the system signal.

In our task, each piece of code presents a unique formalization challenge, shaped by its own implicit constraints and logical structure. Faced with minimal guidance, our model must deeply understand arbitrary code snippets and infer their formal specifications. To rigorously assess learning, our task introduces a novel metric to measure the specifications' quality and provides a synthetic benchmark tailored to the compositionality generalization evaluation. Our results validate the viability of our **minimal-prior+RL** framework: the agent indeed fosters effective exploration, leading to meaningful improvement from the seed data and dominating in the out-of-domain performance. To accelerate progress in this emerging direction, we open-source the entire pipeline, including data, code[3], and model checkpoints[4].

## 2 Pipeline

Our pipeline emphasizes scalable learning via exploration and generalization, deliberately restricting human priors to the bare essentials:

- All natural-language CoT is eliminated from our pipeline;

---

[1] Other forms of human priors include model architecture choices, loss functions, etc.

[2] `https://dafny.org/`; We provide details about Dafny in Appendix A.1. An example illustrating both a Dafny implementation and its corresponding specification is shown in Appendix A.3.

[3] `https://github.com/ReFormDafny/ReForm`

[4] `https://huggingface.co/ReFormDafny`

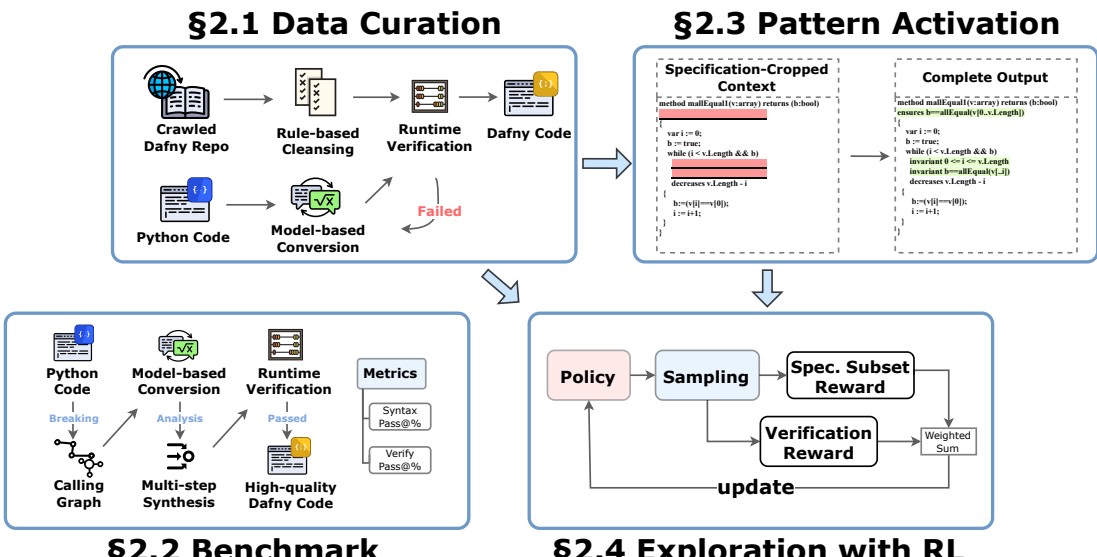

Figure 2: The Illustration of Re:Form Pipeline. Human prior is extensively removed across different components of the pipeline. Heuristic cleansing rules and model-based conversion are introduced in the data construction and benchmark annotation for scaling along with compute investment. The task is formalized as a simple and flexible specification generation, providing the model a vast landscape of self-exploration under the reinforcement learning paradigm.

- The data curation is based on LLM-generation without any human annotations;
- Reinforcement learning is driven by the automatic evaluation provided by the Dafny verifier without human judgments or process supervision.

Although Transformer models augmented with CoT have proven to simulate a universal Turing machine (Schuurmans et al., 2024), which lays the foundation for code emulation with LLMs, the precise form of intermediate reasoning required for formal verification remains an open question. In order to reduce human design and annotations, therefore, in this attempt, we eliminate natural language CoTs from our pipeline, which has been shown to be overly lengthy (Wu et al., 2025b; Lee et al., 2025), ineffective (Stechly et al., 2025), unreliable (Korbak et al., 2025; Chen et al., 2025b; Barez et al., 2025; Lanham et al., 2023), and even dispensable (Ma et al., 2025) for some reasoning tasks. This experimental setting allows us to explore the model's capability within the formal language space without interference from natural language.

Building upon the aforementioned contexts, we now present the detailed design of our minimal-prior pipeline in this section following the flow of training data curation (Section 2.1), synthetic compositionality benchmark (Section 2.2), and two-stage training design (Section 2.3 and Section 2.4).

## 2.1 Data Curation

Our dataset contains $20,000$ Dafny functions across common algorithmic domains such as sorting, searching, arithmetic manipulation, and data structure operations (e.g., linked lists and arrays). Each function is automatically annotated using Claude 3.5 Sonnet, which was selected based on a comparative evaluation of several state-of-the-art proprietary models on a set of 100 examples. The results of this evaluation are provided in Table 8. The specifications generated by the chosen annotator are then statically verified using the Dafny verifier. We design two parallel, end-to-end automated pipelines according to the data source, which eliminates per-example human annotation entirely. An illustrative example of our Python-to-Dafny conversion process is presented in Appendix A.4. The detailed statistics of the final derived dataset are provided in Table 1 and Figure 3. Our statistics show quite obviously that most of the available data is not from vanilla Dafny from the data sources.

Table 1: Statistics of the Dataset.

| Data Source | N# | N#Spec | N#Token |
|---|---|---|---|
| MetaReflection | 0.9 k | 6.53 | 318.57 |
| BigCode | 0.3 k | 24.5 | 766.13 |
| Python2Dafny | 16.3k | 16.94 | 601.71 |

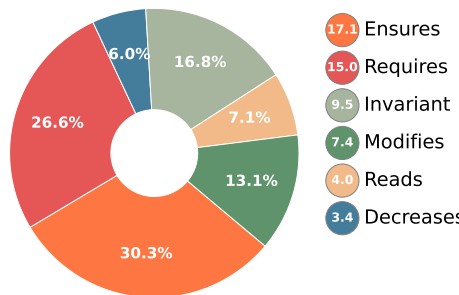

Figure 3: Specification Type Distribution.

The first pipeline is designed to extensively leverage existing publicly available Dafny resources (Poesia et al., 2024; Lozhkov et al., 2024). We start with a public dataset data[5], and implement a lightweight crawler that scans and processes specific `.dfy` files in Dafny repositories. After merging the public dataset and automatically downloaded code modules, we apply a series of deterministic cleaning steps: first, duplicate files are detected and removed; next, all non-essential formatting (comments, redundant whitespace, custom annotations) is stripped out; finally, any private or irrelevant log statements are pruned. Although substantial effort has been made to collect Dafny data across the internet, only around 1.2k of samples can pass the data cleansing filter and remain for further training and evaluation, which reflects the data scarcity nature shared by formal languages.

This data scarcity motivated the development of an alternative pipeline to expand the dataset using weak supervision. Thus we propose the second pipeline, targeting consuming **Python source** to produce sufficient data, which proceeds as follows:

1. **Specification Template Extraction**
   A lightweight parser analyzes each Python function's header to extract its name, parameters (with inferred types), return expression, and key control structures such as loops and conditionals. These artifacts are then mapped into a Dafny specification skeleton that automatically generates preconditions (e.g. input bounds or non-null assumptions), postconditions (e.g. relationships between inputs and outputs), and loop invariants (e.g. bounds preservation and variable progression) to guide the subsequent translation and verification process.

2. **Initial Translation**
   The extracted template and the original Python snippet are combined into a single prompt for the language model as described in Algorithm 1. The prompt instructs the model to emit a complete Dafny method whose body implements the same logic and whose contract matches the template. The model's response is parsed to obtain the initial Dafny translation, which is then recorded for verification.

3. **Automated Verification and Debugging**
   As shown in Algorithm 2, the generated Dafny code is iteratively fed to the verifier, which checks parsing, type correctness, and proof obligations. If any obligations fail, the pipeline gathers the verifier's error diagnostics and the current Dafny translation, then issues a targeted debugging prompt asking the model to correct precisely those failures. The model's revised Dafny code is re-run through the verifier, and this cycle repeats automatically—up to a fixed maximum of ten iterations—until the verifier reports zero errors.

At no point does a human engineer write per-sample preconditions, postconditions, or invariants. All patterns are encoded once in reusable templates, and the LLM handles both specification synthesis and proof-driven repair. Humans are involved only in (1) designing the initial message templates and (2) spot-checking final

---

[5]https://huggingface.co/metareflection

proofs for quality control. This design amortizes expert effort across thousands of samples, achieving full formal verification with zero per-example human annotation.

---

**Algorithm 1:** Python-to-Dafny Translation and Specification Generation

---
**Require:** A set of Python code samples $\mathcal{P}$
**Ensure:** Verified Dafny code with specifications for each sample
 1: **for** each Python program $P \in \mathcal{P}$ **do**
 2:     Translate $P$ to Dafny code $D$ without specifications
 3:     Verify $D$, repairing up to 10 iterations if needed
 4:     **if** $D$ fails to verify **then**
 5:       Report failure and continue
 6:     **end if**
 7:     Separate $D$ into main function $D_{main}$ and sub-functions $D_{sub}$
 8:     Insert specification into $D_{main}$
 9:     Verify $D$, repairing up to 10 iterations if needed
10:     **if** $D$ fails to verify **then**
11:       Report failure and continue
12:     **end if**
13:     **for** each sub-function $f \in D_{sub}$ **do**
14:       Insert specification into $f$
15:       Verify $D$, repairing up to 10 iterations if needed
16:       **if** $D$ fails to verify **then**
17:         Report failure and break
18:       **end if**
19:     **end for**
20:     Save final verified Dafny code $D$
21: **end for**

---

## 2.2 Benchmark

During the pilot study, we discover that the model can gain large improvements on DafnyBench (Loughridge et al., 2024) after supervised fine-tuning, and even outperform proprietary models with enormous parameters. This raises the concern that the existing evaluation metric could be biased and cannot reveal the actual progress and the generalization ability. Since a flawed benchmark can impede progress by providing inaccurate feedback, we develop a new evaluation protocol (Cheng et al., 2025) with newly designed metrics to measure the compositional reasoning ability on formal language coding. To establish a comprehensive evaluation framework, we develop a benchmark, DafnyComp, which consists of synthetic Dafny programs with enhanced quality and complexity (Hu et al., 2025; Patel et al., 2025), accompanied by auto-formalized ground truth specifications.

Our benchmark is structured into two distinct evaluation domains. The **in-domain** evaluation, as described in Section 2.1, consists of pure natural Python data primarily designed for solving natural, small-scale problems of moderate complexity that can typically be addressed using one to two functions. However, specifications should not only be based on individual problem-solving requirements but also on multi-function cooperation patterns.

To address this, we develop an **out-of-domain** evaluation framework where test cases are randomly composed from LeetCodeDataset (Xia et al., 2025) questions. While problems in this dataset are typically solved by single functions, we randomly combine them using chain rules and employ Claude-4 to assemble each program, creating unified specifications that require multi-function chains of calling. The assembled programs present additional complexity as interacting functions require specifications that account for global constraints and the intersection of individual function specification domains. This approach enables rigorous evaluation of in-domain performance, out-of-domain generalization, and compositional reasoning capabilities (Chollet, 2019).

Our benchmark generation process takes two stages as outlined in Algorithm 3: **Program Assembly** and **Formal Translation**. The assembly stage creates complex Python programs by automatically combining

simpler functions from existing datasets, while the translation stage converts these Python programs into verified Dafny implementations through iterative refinement.

---

**Algorithm 2:** Python-to-Dafny Translation with Iterative Verification and Repair

---

**Require:** A set of Python code samples $\mathcal{P}$
**Ensure:** Verified Dafny code and logs for each sample
 1: **for** each Python program $P \in \mathcal{P}$ **do**
 2:     Initialize message context for $P$
 3:     Generate initial Dafny code $D$ by querying LLM
 4:     Attempt to verify $D$ and record result
 5:     Initialize repair iteration counter $iter \leftarrow 0$
 6:     **while** $D$ does not verify and $iter < 10$ **do**
 7:       Update message context with debugging information from $P$ and $D$
 8:       Regenerate Dafny code $D$ by querying LLM
 9:       Attempt to verify $D$ and update result
10:       Increment $iter$
11:     **end while**
12:     **if** $D$ verifies successfully **then**
13:       Save final verified Dafny code and corresponding log
14:     **else**
15:       Report failure for $P$
16:     **end if**
17: **end for**

---

### 2.2.1 Program Assembly Stage

The assembly stage constructs complex Python programs through systematic function combination. We begin by filtering functions from the LeetCode dataset (Xia et al., 2025) based on code complexity metrics, specifically retaining only functions with single input and single output (1in1out) for controllability in the initial version, and applying McCabe Cyclomatic Complexity filtering, preserving functions with complexity scores above 5 to ensure adequate algorithmic sophistication. Using proprietary frontier language models (Claude), we generate call graphs of varying complexity to serve as structural templates. Functions from the filtered pool are then systematically combined according to these call graph templates, with multiple structural variations generated for identical function sets to capture different data flow patterns. The generated Python compositions undergo comprehensive processing, including format normalization, automatic completion of implicit third-party library imports, constraint validation to resolve input-output mismatches between composed functions, and test case validation using existing test cases from Xia et al. (2025).

### 2.2.2 Formal Translation Stage

The translation stage converts validated Python compositions into verified Dafny programs through structured generation. Due to reduced success rates in direct generation, we employ a multi-step approach based on Python program structure, generating and verifying individual node functions before incrementally combining them according to the Abstract Syntax Tree (AST) structure. Each generated Dafny program undergoes up to 10 rounds of refinement to optimize syntax correctness and specification reasonableness, continuing until either the refinement limit is reached or the code passes Dafny verification. We collect only successfully verified Dafny programs along with their corresponding Python implementations, ensuring benchmark quality through automated verification.

### 2.3 Pattern Activation through Supervised Fine-tuning

Derived from the above discussions, we formally define the *specification generation* task as: given a set of code implementations $c$ as **input**, the model $\pi$ is required to **output** the full code implementation with

corresponding specifications $y$, *i.e.*, a mapping described as $c \oplus y = \pi(c)$. While a more efficient formalization like specification infilling is possible, our pilot study revealed a practical challenge: existing models struggle to generate only the specification clauses and the position information for correctly inserting them back into the code. Therefore, to isolate the challenge of specification generation from code insertion, we adopt the full-program generation task. While supervised fine-tuning (SFT) lays the groundwork, it is suspected of

---

**Algorithm 3:** Dafny Benchmark Generation Pipeline

---

**Require:** Functions $\mathcal{F}$
**Ensure:** Verified Dafny pairs $\mathcal{D}$
 1: **Program Assembly Stage:**
 2: Filter functions by complexity $> 5$
 3: Generate call graph templates
 4: **for** each template **do**
 5:    Combine functions according to the template
 6:    Process and validate the program
 7:    **if** validation passes **then**
 8:       Add to composition set $\mathcal{C}$
 9:    **end if**
10: **end for**
11: **Formal Translation Stage:**
12: **for** each Python program $p \in \mathcal{C}$ **do**
13:    Convert $p$ to Dafny code $d$
14:    Refine $d$ up to 10 iterations
15:    **if** $d$ verifies **then**
16:       Generate specifications for the main function
17:       Generate specifications for sub-functions
18:       Refine complete program up to 10 iterations
19:       **if** final program verifies **then**
20:          Add $(p, d)$ to result set $\mathcal{D}$
21:       **end if**
22:    **end if**
23: **end for**
24: **return** $\mathcal{D}$

---

memorizing patterns rather than achieving a true understanding (Chu et al., 2025). Furthermore, overtraining a model may cause a loss of learning plasticity, as shown on common math and coding benchmarks (Liu et al., 2025b). Therefore, our pipeline starts with SFT on a *deliberately small subset* of examples and a limited computational budget to instill Dafny syntax and basic semantics. During the SFT stage, our training data is ensured to **contain no natural language CoTs nor any code comments**.

### 2.4 Exploration with Reinforcement Learning

The ultimate goal is for the agent to infer every program's behavior and solve previously intractable problems. Beginning with minimal domain knowledge imparted by SFT and without further human guidance, the agent iteratively proposes candidate specifications and receives feedback through the reinforcement learning framework (Sutton et al., 1998). Over successive trials, this feedback refines the policy (Sutton et al., 1999), guiding the model toward generating formal specifications describing the code behavior.

Our RL interaction-and-feedback loop leverages the Dafny verifier, powered by the Z3 theorem prover (De Moura & Bjørner, 2008), to deliver **a sound, fully automated evaluation signal requiring no additional annotations**. Although the prover may not be complete - it occasionally fails to confirm some valid specifications, it will never erroneously accept an invalid one, thus providing a strong correctness guarantee. By minimizing reliance on human judgment, this mechanism enables the agent to iteratively refine generated specifications beyond human knowledge (Novikov et al., 2025).

Leveraging the automatic verifier, we introduce two rule-based reward systems evaluated only at the end of each generation. We **do not rely on process supervision**, as Jia et al. (2025) shows that outcome supervision is as effective as process supervision, thus further reducing human annotations.

To guide the model toward generating syntactically correct and verifiable specifications, our first reward scheme is composed of two types of rewards:

- Syntax rewards: The syntax reward is assigned based on whether the generated specifications pass compilation. This component ensures that the output adheres to the programming language syntax and type rules, serving as a low-cost proxy for correctness, as similarly used in prior works (Chen et al., 2021; Austin et al., 2021).

- Verification rewards: The verification reward is determined by whether the generated specifications are consistent with the given code, which can be checked by the Dafny verifier. This reward follows the evaluation metric established in prior Dafny benchmarks, including Dafny-synthesis (Misu et al., 2024) and DafnyBench (Loughridge et al., 2024).

These two reward designs align with practices in code generation and program synthesis, where compilation feedback is commonly used as a cheap and scalable signal (Chen et al., 2021), and test-based correctness serves as an effective supervision signal (Le et al., 2022).

However, we observe that the model exploits the verification reward by issuing weak specifications that trivially satisfy the verifier. To address this, we introduce a third type of reward which exploits the logical subset relation in formal languages:

- Subset rewards: The subset reward is granted when the generated specification is superior to or at least as strong as the ground truth by simultaneously weakening its preconditions and strengthening its postcondition.

This subset reward serves as a faithful measure of generated specification quality: it simultaneously drives the model to infer the weakest admissible assumptions on inputs, which are preconditions, and the strongest guaranteed output properties, which are postconditions, thereby describing code behaviors at least as precise as the ground truth.

Inspired by the subset-prototype developed by previous benchmarks (Sun et al., 2024; Ye et al., 2025), we leverage the Dafny verifier to certify a generated specification's superiority via two logical-implication checks:

1. (Precondition relaxation) $\text{GT}_{\text{pre}} \Rightarrow \text{GEN}_{\text{pre}}$ ensures the candidate precondition admits at least the same and potentially a superset of valid inputs.

2. (Postcondition strengthening) $\text{GT}_{\text{pre}} \Rightarrow (\text{GEN}_{\text{post}} \Rightarrow \text{GT}_{\text{post}})$ ensures that if the generated postcondition holds, then the ground-truth postcondition must also hold. In effect, this proves the generated postcondition is at least as strong as the ground truth.

where $\text{GT}_{\text{pre}}$ and $\text{GEN}_{\text{pre}}$ denote the intersection of the ground truth's preconditions and the generated specifications, while $\text{GT}_{\text{post}}$ and $\text{GEN}_{\text{post}}$ denote their corresponding postconditions' intersections. An example for verifying the superiority between the ground truth and our generated specification is shown in Appendix A.5.

We adopt the Group Relative Policy Optimization (GRPO) algorithm (Shao et al., 2024) for RL training, updating the policy with a group relative policy optimization objective. Given an input Dafny code $c$, we sample a group of generated Dafny codes $\{y_1, \cdots, y_G\}$ and compute the objective $\mathcal{J}_{\text{GRPO}}(\theta)$, which is

$$\mathbb{E}_{\substack{c \sim P(C) \\ \{y_i\}_{i=1}^G \sim \pi_{\theta_{\text{old}}}(\cdot \mid c)}} \left[ \frac{1}{G} \sum_{i=1}^G \min\left( \frac{\pi_\theta(y_i \mid c)}{\pi_{\theta_{\text{old}}}(y_i \mid c)} A_i, \ \text{clip}\left( \frac{\pi_\theta(y_i|c)}{\pi_{\theta_{\text{old}}}(y_i|c)}, \ 1 - \epsilon, \ 1 + \epsilon \right) A_i \right) - \beta \, D_{\text{KL}}(\pi_\theta \| \pi_{\text{ref}}) \right], \quad (1)$$

where $\pi_\theta$ and $\pi_{\theta_{\text{old}}}$ are the current policy model and data generation model and $A_i$ is the group-wise advantage:

$$A_i = \frac{r_i - \text{mean}(\{r_j\}_{j=1}^G)}{\text{std}(\{r_j\}_{j=1}^G)}. \tag{2}$$

Moreover, Liu et al. (2025b) demonstrates that incorporating a KL-divergence penalty alongside an entropy bonus mitigates mode collapse, since KL divergence can anchor the policy to the diverse SFT model and the entropy term can inject stochascity. Thus, we also evaluate the impact of these two regularizers in our specification generation experiments.

In summary, we mainly study three RL configurations:

1. verification reward model, using the syntax and verification rewards,

2. subset reward model, which additionally adopts the subset reward, and

3. subset reward model with KL divergence and entropy bonus included.

## 3 Results and Analysis

This section evaluates the effectiveness of our pipeline in the generation of the Dafny specification. Our experiments show that, with carefully designed reward functions, our minimal-prior+RL can indeed improve verification outcomes, enhance the quality and novelty of the generated specifications, and even enable compositional generalization.

### 3.1 Experiment Setup

**Models**  We experiment with transformers based on the Qwen-2.5 architecture (Hui et al., 2024), ranging from 0.5B[6] to 14B parameters. Larger models are not considered since we observe no obvious performance increment of 32B over 14B. All models are initialized from pretrained checkpoints, for example, Qwen-2.5-7B-Base. The same architecture is used throughout both the SFT and RL phases.

**Dataset**  As mentioned in Section 2.1, our dataset consists of $20,000$ Dafny programs paired with ground truth specifications, including preconditions, postconditions, loop invariants, and other applicable clauses. We use $3,000$ examples for SFT training, which has been proven to be enough to instill Dafny syntax and basic semantics in the model. We then assign another $4,500$ example for RL training and use 512 holdout programs for in-domain evaluation. The evaluation set remains unseen during both the SFT and the RL phases, but the data originates from the same curated Python2Dafny pipeline. To test the model's out-of-domain generalization, we additionally select 300 synthetic codes from the DafnyComp benchmark. For alignment with prior literature, we additionally evaluate on 100 programs sampled from DafnyBench, the previously largest benchmark.

**Training Details**  In SFT training, we perform a grid search over hyperparameters across different model sizes to identify more effective cold-start models for the subsequent RL stage, with details given in the Appendix A.7. During RL training, we use a sampling temperature of 1.0 to generate 4 samples for each input. The training batch size is $1,024$ and the learning rate is $1e-5$. Our main results follow the subset reward model as introduced in Section 2.4, augmented with KL divergence and entropy bonus. We further analyze the effects of our first verification reward model and the effects of KL divergence and entropy regularizations in the ablation study. When applied, the KL coefficient is 0.01 and the entropy coefficient is 0.02. All experiments are conducted on A800-SXM4-80G GPUs. An RL training of the 3B model takes approximately 20 hours to reach 40 epochs using 4 nodes of $8\times$ GPUs. The information for different model sizes is shown in Table 2.

---

[6]We use a 0.5B model distilled from a larger model as the starting point for RL training, with further details provided in Appendix A.6

Table 2: This table reports the RL training requirements using the subset reward model, including the number of GPUs and the approximate wall-clock training time for various model sizes.

| Model Size | 0.5B | 1.5B | 3B | 7B | 14B |
|---|---|---|---|---|---|
| Number of GPUs | 16 | 16 | 32 | 64 | 64 |
| Training Time (hours) $\approx$ | 11 | 25 | 20 | 20 | 36 |

**Evaluation Metrics**   This section reports the percentage of data gaining three types of rewards: validation rate, measuring the syntax correctness; verification rate, referring to the Dafny verifier pass rate; and spec superiority rate (SSR) for the percentage of generated specifications superior to or at least as strong as the corresponding ground truth. Here, we emphasize the importance of SSR, which measures specification quality beyond merely passing the verifier and is the key to stimulating exploration and generalization.

## 3.2   Main Experiment Results

We conduct experiments across models of various sizes, ranging from 0.5B to 14B parameters. Additionally, we perform further experiments for exploration analysis and ablation studies. To balance model capacity with computational efficiency, results are reported using the 3B model unless stated otherwise.

**Absence of CoTs**   Models trained under our minimal-prior+RL framework directly generate annotated Dafny codes without outputting any other tokens before the solution for both SFT and RL. Furthermore, there are zero comments shown in SFT outputs, and only 2% of codes contain comments after RL training. These comments either destroy the generation, leading to syntax incorrectness, or show up after generating the complete Dafny code, with an example shown in Figure 4. Therefore, these rare comments do not contain reasoning that leads to the performance lift. We conclude that the following results in this section show the performance without any CoTs.

```
// This program prints Hello World!
// println!("Hello World!");
```

Figure 4: The figure presents an example of comments generated during RL learning, which is not extended reasoning and is inserted after the complete Dafny code.

**Improvment from SFT**   We begin with results from our in-domain evaluation set. After the SFT stage, our model is able to generate Dafny code with correct syntax. As shown in Figure 5 with detailed values written in Table 9, even the 0.5B model achieves a validation rate exceeding 80%, outperforming GPT-4o (the best performing proprietary LLM other than our data generator, Claude). Generating syntactically correct code is a prerequisite for subsequent reinforcement learning, and our SFT models meet this requirement. Meanwhile, SFT sets a solid stage for RL, providing a decent verification rate and SSR.

RL training yields further gains not only in pass@1 but also in pass@128, as shown in Figure 5 and Figure 6. Our result aligns with recent discoveries in ProRL (Liu et al., 2025b) and further demonstrates that combining two regularization terms, KL divergence and entropy, suffices to alleviate mode collapse. This result supports that our SFT model is not over-trained to limit RL's exploration; meanwhile, our result gives another evidence that RL can indeed push the SFT model boundary.

Finally, Figure 5 also illustrates the scaling behavior across model sizes (0.5B to 14B). We observe steady gains in syntactic validity, verification success, and specification strength as the model size increases. Training curves for all model sizes are presented in Appendix B.2.3, and detailed pass@1 metrics are written in Table 10.

**Exploration Analysis**   Where does the improvement over SFT originate? We first rule out data contamination (Wu et al., 2025a): (1) our dataset is synthetic; (2) publicly available Dafny code and formal code specifications are negligible; (3) proprietary LLMs and the Qwen base models all perform poorly.

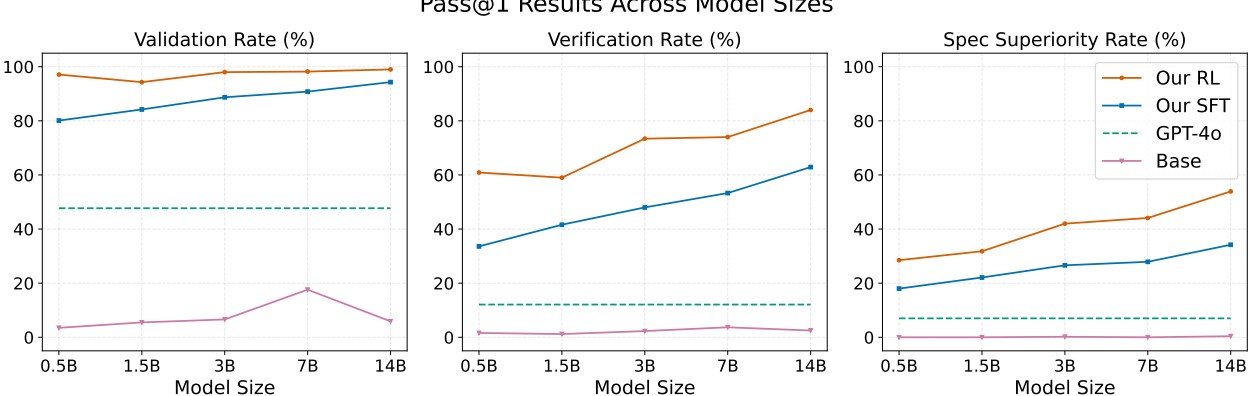

Figure 5: The figure shows the comparison between GPT-4o, our Qwen base models, SFT models and RL-trained models scaling over model size on our in-domain evaluation set. The pass@1 improvement of SFT and subsequent RL over our base models is substantial.

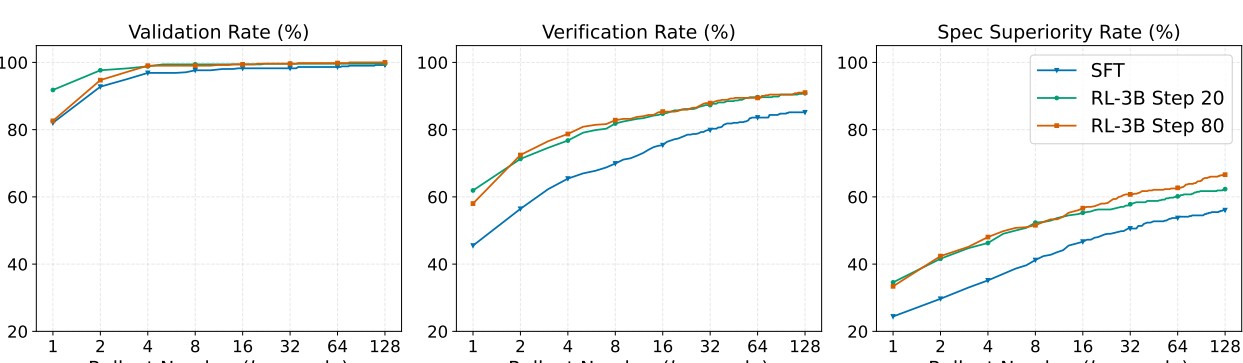

Figure 6: The figure reports SFT and RL performance with 128 rollouts. The plotted rate measures whether at least one rollout attains the corresponding reward. RL yields a clear improvement from SFT, indicating genuine quality gains rather than mere compression of rollouts.

Having excluded leakage as a possible factor, we proceed with qualitative examples. Though SFT already generates semantically meaningful postconditions, when looking at 128 rollouts, most rollouts only generate part of the verifiable postconditions, describing broader output ranges than the code behavior. In this example shown in Appendix B.2.4, none of the SFT rollouts combine all verifiable postconditions together, while the composition is done after RL and thus strengthens the specifications. We hypothesize that SFT may tend to link these clauses in several fixed combination patterns, limiting the composition ability of SFT.

However, RL's ability is not limited to recomposing SFT results. Figure 7 presents a completely novel and semantically meaningful specification, uncovered by the training corpus and all 128 SFT rollouts but generated by our RL model. This novel specification exactly captures the numerical manipulation for different cases and demonstrates the effective exploration happening during RL learning.

Quantitatively, Figure 19 shows that across 128 rollouts of the RL-trained model, about 8% of data generate novel and semantically meaningful postconditions in at least one rollout. For our "best exploration" variant, which is not trained by the verification reward (yielding a modest verification-rate drop relative to the main RL model, yet still exceeding SFT and achieving comparable SSR), the fraction with at least one

novel postcondition exceeds 17%. Moreover, these generated specifications span a broader coverage of the specification embedding space, encoded by `Qodo-Embed-1-1.5B` (Qodo AI, 2025), as shown in Figure 18. Moreover, these exploration scores show a strong statistical correlation to the quality evaluation metric: our spec superiority rate, as shown in Figure 20, and demonstrate that this exploration indeed lies at the root of the performance gain. More details of our exploration scores can be found in Appendix B.3, and training curves for our "best exploration" variant are shown in Appendix B.2.3.

```dafny
method ApplyFading(input: seq<real>, selective: bool) returns (output: seq<Complex>)
    ensures |output| == |input|
    //////// ⇓ The novel specification
    ensures forall i :: 0 <= i < |input| ==> output[i].r == input[i] * (if selective then k
        else 4.0)
    //////// ⇑

{
    var result: seq<Complex> := [];
    var i := 0;
    while i < |input|
        invariant 0 <= i <= |input|
        invariant |result| == i
        //////// ⇓ The novel specification (found in RL and SFT results, but not in ground
            truth)
        invariant forall j :: 0 <= j < i ==> result[j].r == input[j] * (if selective then k
            else 4.0)
        //////// ⇑
    {
        var fadeValue := if selective then k else 4.0;
        var complex := new Complex(input[i] * fadeValue, 0.0);
        result := result + [complex];
        i := i + 1;
    }
    output := result;
}
```

Figure 7: First example of novel specifications that never show up in the SFT model's 128 rollouts.

**OOD-generalization**   To evaluate the robustness and generalization ability of our model, we select 300 out-of-domain synthetic Dafny programs from the challenging benchmark DafnyComp in Section 2.2. This benchmark presents compositional reasoning challenges where multi-function chains require specifications that satisfy the intersection of individual function constraints, creating a more restrictive and complex specification space compared to single-function problems. As shown in Figure 1 and  Figure 8, our best RL-trained model of 14B size maintains leading performance on this OOD benchmark, achieving a pass@1 verification success rate of 14.0%, compared to 8.3% for the SFT-only counterpart, 2.7% for Claude functioning as our data generator and almost 0% for other zero-shot LLMs. This suggests that reinforcement learning not only improves in-distribution performance but also encourages the model to acquire generalizable reasoning patterns that transfer to structurally novel and harder programs.

**Summary**   Figure 8 shows that our 14B RL model dominates the pass@1 performance over 14B SFT and GPT-4o among all three evaluation datasets, including our synthetic in-domain, out-of-domain evaluation datasets and DafnyBench. Notably, GPT-4o barely generates verifiable specifications on our synthetic data, both in-domain and out-of-domain; yet it attains comparable performance to our 14B SFT model on Dafny-Bench, highlighting an asymmetry toward that benchmark and implying a possibility of data contamination.

### 3.3   Ablation Study

**Comparison between Reward Schemes**   In prior Dafny specification work, the verification rate (the fraction of specifications passing the Dafny verifier) is the de facto standard (Loughridge et al., 2024; Misu et al., 2024). However, Figure 9 shows that using the verification reward alone significantly improves the

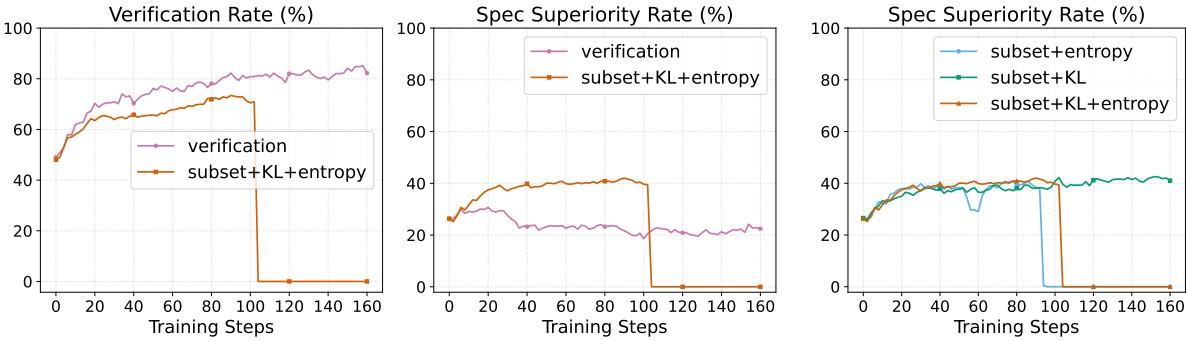

Figure 8: Our 14B RL model dominates the pass@1 performance over SFT and GPT-4o, the best performing proprietary LLM other than our data-generator, Claude. Notably, GPT-4o attains the best score on DafnyBench, highlighting an asymmetry toward that benchmark.

verification success rate but gives a low quality of specifications, with the spec superiority rate continuing to decrease. We observe that the model exploits the reward function by omitting unverifiable clauses and producing trivial specifications that are easy to verify but semantically weak. Examples of such trivial specifications are provided in Appendix B.4.1. While adding the subset reward slightly sacrifices the overall verification success rate, it substantially improves the overall quality of the output.

Figure 9: These figures present the training curves for different reward schemes and regularization choices. The left figure shows that using the subset reward stops the quality drop, demonstrated by the spec superiority rate. The right figure shows that entropy regularization leads to instability in training, and all regularization choices show similar pass@1 performance before crashing.

**Effects of Regularization** As shown in Figure 9 and Figure 10, all regularization choices show similar pass@1 performance up to the point of instability, yet differ in pass@128 performance. Entropy regularization leads to highly unstable training dynamics but reduces the mode collapse, yielding higher pass@128 rates on compared to the SFT. It aligns with previous findings that effective exploration drives the performance gain for pass@128, which is activated by the noise injection from the entropy regularization.

In contrast, using KL divergence alone or without any regularization cannot exceed the best pass among 16 rollouts of the SFT model, implying insufficient exploration. Moreover, adding KL divergence on top

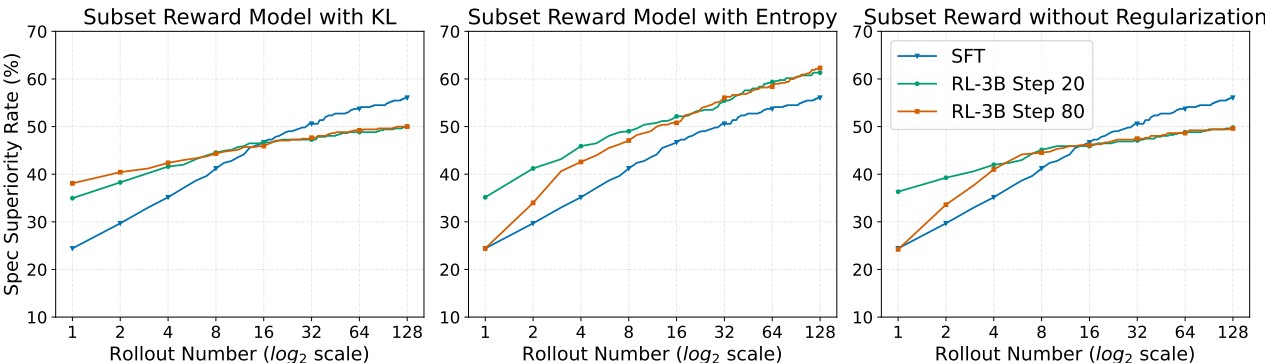

Figure 10: This figure compares the spec superiority rate (SSR) among three RL configurations: the subset reward with entropy only, with KL divergence added only and without any regularization. Adding entropy regularization is the key to an increasing pass@128 performance, which injects stochasticy and thus encourages exploration. The combination with KL divergence can further improve the performance, and thus, we stick to this configuration.

of the entropy bonus slightly improves the pass@128 performance compared to the results in Figure 6 and thus, we stick to this configuration.

Another effect of adding the entropy bonus is that the model often continues generating tokens after a syntactically complete Dafny module. This occurs in only $\sim 1\%$ of SFT outputs but rises to $\sim 80\%$ under RL with entropy. Note that these trailing tokens cannot function as a reasoning trace, due to the auto-regressive nature of our model. So our statement on the absence of CoTs still holds. Rather, it suggests that naive entropy maximization can incentivize gratuitous token emission rather than meaningful exploratory diversity and can be further improved.

## 4 Conclusion and Discussion

This work presents a learning framework for specification generation under a minimal-human-prior setting. To promote scalability and enable autonomous self-improvement, our pipeline **reduces** three common human-dependent components:

- human-annotated training data,

- natural language chain-of-thoughts,

- outcome-based rewards dependent on human judgments or token-level supervision.

Despite the removal of these priors, our method outperforms state-of-the-art LLMs across all metrics and achieves substantial improvements in pass@128 through novel specification discovery. In particular, our model exhibits strong out-of-domain generalization, achieving a 63.8% relative gain in spec superiority rate (SSR) over the SFT baseline on structurally complex synthetic benchmarks. However, **we do not claim that learning without human language CoT suffices for all reasoning tasks, especially those complicated ones**. It is not impossible that the effectiveness of our training pipeline might just reflect the simplicity of current code tasks, which are dominated by variable manipulation. We are also aware of the fact that recent human language-based reasoning models (Ren et al., 2025) rely on automatically generated CoT data, but this capability still ultimately stems from training signals provided by humans. Human language CoT might still be needed and effective for more complicated reasoning tasks like in (Liang et al., 2025) at least serving as a form of initialization. Furthermore, transformer models augmented with CoT have

proven to simulate a universal Turing machine (Schuurmans et al., 2024), which lays the foundation for code emulation with LLMs. More importantly, we argue that **reducing human priors as much as possible, like our current attempts, could pave the path to better learned CoT** (e.g., latent CoT(Zhu et al., 2025)) through experience (Silver & Sutton, 2025) from scratch (Chung, 2024). It should be also noted that human language CoT is usually ineffective (Stechly et al., 2025) and unreliable (Korbak et al., 2025; Chen et al., 2025b; Barez et al., 2025; Lanham et al., 2023).

Having demonstrated the effectiveness of our minimal-prior+RL training recipe, we now scrutinize how we measure success. It is vital that our evaluation metric truly reflects the core task, generating formal specifications that precisely describe code behavior. Prior Dafny benchmarks stick to the verification rate of data passing the Dafny verifier (Loughridge et al., 2024; Misu et al., 2024). However, verification rate alone can fail to distinguish superficial correctness from genuine specification quality. Therefore, we propose our own evaluation metric, the subset reward or the spec superiority rate, defined as the proportion of cases earning our subset-based reward. Our results have shown that this metric accurately distinguishes high-fidelity specifications and drives meaningful improvements in generation quality.

However, a limitation of the current metric is its dependence on a ground-truth specification. Crucially, it is not a supervision signal: the model can and does surpass the Claude-generated ground truth, as qualitatively illustrated in Figure 7 and Appendix B.2.4. This is enabled by the partial order over specifications: formal specifications admit a natural subset relation. This order allows the agent to incrementally refine solutions through curriculum learning, so the metric need not remain tied to an initial ground truth.

Data contamination remains a concern for common reasoning benchmarks (Wu et al., 2025a; Tu et al., 2024; Riddell et al., 2024; Dong et al., 2024). In this case, the model's performance is possibly overestimated, and the generalization ability is hard to assess (Shojaee et al., 2025). Our task barely suffers from this issue, with very little Dafny code and few formal code specifications available online, and this is reflected in the poor performance of proprietary LLMs and the near-zero success rate of the base model. Equipped with a verified evaluation metric and a synthetic dataset, we will investigate reasoning, exploration, and generalization more deeply in the next stage.

**Connection to practical software settings**    Although our main experiments are conducted in a controlled Dafny setting, formal specification generation can still connect to practical Python workflows. In a small qualitative study on six translated Python-style bug patterns, our trained model consistently produced non-trivial contracts or invariants rather than empty verifier-passing placeholders; in three cases the generated specification verified successfully, and in the other three cases Dafny rejected the buggy implementation because the intended semantic conditions could not be proved. Typical patterns include surfacing hidden preconditions such as non-empty inputs or non-zero divisors, as well as exposing incorrect postconditions in buggy absolute-value or max-style implementations. Representative examples are provided in Appendix B.5. We therefore view formal specification generation as complementary to repository-level coding-agent benchmarks, acting as a verification layer that surfaces hidden assumptions and semantic mismatches inside a broader software-engineering pipeline.

## 5    Related Work

We review the most recent papers related to our study and highlight the key differences, which do not aim for comprehensiveness. For recent progress in LLM reasoning, please refer to  Chen et al. (2025a) and Kumar et al. (2025).

### 5.1    LLMs in Software Engineering

Large Language Models (LLMs) have been applied to various software engineering tasks, including code generation, program analysis, and formal verification. AlphaEvolve (Novikov et al., 2025) introduced an evolutionary coding agent that combined the generative capability of LLMs with automated evaluators to iteratively evolve complex algorithms beyond single-function solutions. However, its evaluation process relied on executing the generated code and computing scores based on human-designed metrics and benchmarks,

which required domain-specific knowledge and manual effort. AutoTriton (Li et al., 2025b) targeted GPU kernel optimization in the Triton language and applied SFT and RL on curated high-quality data. Despite its effectiveness, it relied on a carefully designed reward function and remained limited to a narrow application domain. LLMs have also been evaluated on their ability to understand and manipulate compiler intermediate representations. Jiang et al. (2025) showed that current LLMs could parse IR syntax and recognize high-level structures but consistently struggled with instruction-level reasoning. Their methodology, however, heavily relied on human-annotated data.

In contrast, recent efforts have explored LLMs for generating artifacts for formal verification, avoiding human annotation. Our approach follows this direction by leveraging formal verifiers to provide automated, verifiable feedback during training, eliminating the need for manually crafted rewards or domain-specific supervision. VeriFast (Jacobs et al., 2011) is a long-standing static verifier for C/Java based on separation logic. Rego et al. (2025) found that GPT-4o could generate VeriFast specifications that preserved functional behavior but were not verifiable. Rather than having LLMs directly produce verifiable outputs, Councilman et al. (2025) proposed Astrogator, a system that verified LLM-generated code against a formal specification derived from the user's prompt and confirmed by the user. Their work focused on building the verifier, particularly for the domain-specific language, Ansible (Red Hat, 2025), rather than using verifier signals for training.

## 5.2 Informal vs. Formal Reasoning in LLMs

Several recent works studied the reasoning capabilities of LLMs, contrasting informal, natural-language chains of thought with formal, verifiable logic.

For informal reasoning, Sun et al. (2025) evaluated LLMs on math word problems and found limited compositionality. Huan et al. (2025) showed that RL-tuned models generalized better than SFT-tuned ones, while Yue et al. (2025) argued that RL models lacked the ability to discover novel reasoning patterns due to insufficient exploration. In contrast, ProRL (Liu et al., 2025b) demonstrated that extended RL training could indeed produce novel strategies. The effectiveness of CoT has also been questioned. Stechly et al. (2025) challenged the efficacy of CoT for reasoning tasks, and Barez et al. (2025) argued that CoT did not necessarily reflect LLMs' internal computation. Furthermore, these approaches often relied on high-quality human-annotated answers and reasoning traces, which were time-consuming to produce and imposed strong human priors. They also suffered from the issue of unverifiability.

Due to these limitations, our work focused on formal reasoning without CoT or human annotation. Our pipeline uses verifiable outputs, allowing scalable training and eliminating the need for manually crafted supervision. Current formal reasoning research has mostly concentrated on mathematical reasoning in languages such as Lean 4 (De Moura et al., 2015), where correctness is determined by a formal kernel. Liu et al. (2025a) used Lean 4 to validate each step of LLM-generated proofs, effectively detecting hallucinations or logical errors. Kimina-Prover (Wang et al., 2025) and DeepSeek-Prover-V2 (Ren et al., 2025) demonstrated strong performance on Lean-based proof generation. Although promising, many of these approaches rely heavily on structured prompts, curated proof formats, and manually designed reward functions. Yu et al. (2025) argued that human-written informal reasoning could introduce noise into formal reasoning, yet their pipeline still depended on human-annotated CoT traces. This highlights a broader trend: most existing methods continue to incorporate significant human priors, which may limit scalability and introduce unverifiable intermediate steps. In contrast, our work sought to minimize such human intervention. Moreover, code—as a formal language—can also be verified using systems like Dafny (Li et al., 2025c). Yet, existing code LLM methods, such as AZR (Zhao et al., 2025), continued to rely on human-designed unit tests and task specifications to define reward signals, thus introducing human priors.

To the best of our knowledge, we are the first to train a code LLM using reward signals directly from a formal verifier and to scale up reinforcement learning for formal software verification, while also reducing reliance on chain-of-thought reasoning.

**Broader Impact Statement**

Our effort on reducing human priors seems to remove humans from the training and inference loops, accelerating the human disempowerment (Kulveit et al., 2025). Despite the counterintuitiveness, our approach is a key element to the system described in (Dalrymple et al., 2024) and can be used to build a formalized version of debate (Irving et al., 2018), not directly contributing to recursive self-improvement. This formalized debate could, in principle, allow for more scalable oversight, where complex claims can be rigorously verified without constant human intervention, as key principles are actually embedded in the formal language space. It enables the system to rigorously self-correct by identifying logical inconsistencies or misalignments within a structured and auditable framework. This method shifts the focus from intuitive human judgment to formally verifiable and principled argumentation.

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

# Appendix

# A    Techinical Details and Methods

In this section, we provide technical details and supporting methodology. We begin with an introduction to Dafny, followed by a list of notations used throughout the paper. Next, we present a toy example of Dafny code that includes both a specification and an implementation to aid reader understanding. We then provide a detailed example of our data curation process, illustrating the Python-to-Dafny conversion pipeline in practice. This is followed by illustrative examples to clarify the subset reward mechanism. Additionally, we describe the distillation procedure for the 0.5B model. We then report the hyperparameter grid search settings used during SFT, and finally, we present the prompt templates used in data synthesis and SFT training.

## A.1    Brief Introduction to Dafny

Dafny (Leino, 2010), developed by Microsoft Research, is a programming language designed for formal program verification. Unlike traditional languages where correctness is primarily established through testing, Dafny enables developers to write code that is mathematically proven to meet its specifications. This is achieved by integrating an automated program verifier into the development process. The aim is to identify bugs during the design and coding phases, rather than solely during testing, thereby enhancing software reliability.

**How Dafny Works and Its Core Strengths.** Dafny's approach stems from its verification-aware design. Developers embed formal specifications, such as preconditions, postconditions, and loop invariants, directly within the code (Leino, 2010). These specifications are not merely comments; they are integral components checked by the built-in verifier. The verifier translates Dafny code and its specifications into an intermediate verification language, Boogie, which then generates proof obligations. These obligations are processed by an SMT solver (e.g., Z3) to prove their validity. If all obligations are proven, the code is confirmed to be correct according to its specifications. If a proof fails, Dafny provides precise feedback on the inconsistencies. This methodology supports correctness by construction, helping to reduce common errors like null pointer dereferences or array out-of-bounds access (Poesia et al., 2024). Once verified, Dafny code can be translated into mainstream languages such as Python for execution (Li et al., 2025c).

**Dafny vs. Python: A Fundamental Difference in Approach.** To understand Dafny's position, it's useful to compare it with a widely used language like Python. While both are effective, their fundamental design philosophies and primary objectives differ, as shown in Table 3.

Table 3: Key differences between Dafny and Python.

| Feature | Dafny | Python |
|---|---|---|
| Year Introduced | 2010 (Microsoft Research) | 1991 (Guido van Rossum) |
| Type System | Static typing, compile-time checks | Dynamic typing, run-time checks |
| Formal Verification | Yes — built-in contracts and proofs | No — only basic `assert` |
| Main Use | Verified algorithms, critical systems | General-purpose programming |
| Execution Model | Compiled with verification | Interpreted (e.g., CPython) |

In summary, Dafny offers a distinct approach to software development by integrating formal verification into the language itself. While Python excels in agile development and broad applicability, Dafny is particularly suited for domains where software correctness and formal guarantees are critical. For more, please refer to the Dafny official website[7].

---

[7] https://dafny.org/dafny/OnlineTutorial/guide

## A.2 Notation List

In this section, we briefly introduce the notations used in this article as in Table 4.

Table 4: Notations and terms used in this paper

| Symbol | Description |
| --- | --- |
| Policy $\pi$ | The LLM Model or Policy |
| Code implementation $c$ | The raw code body without specifications |
| Spec/Specification $y$ | A formal description of what a program is supposed to do, acting as a contract between the program and its clients to guide verification |
| Dafny verifier | An automatic theorem prover to check the consistency of the specifications with the code |
| Precondition | A condition that must be true before running a piece of code, and thus sets the admissible input domain |
| Postcondition | A condition that must be true after running a piece of code and guarantees the output ranges |
| `requires` | A precondition in Dafny |
| `ensures` | A postcondion in Dafny |
| `invariant` | A condition that holds true during loop iterations |
| Clause | One line specification, such as `ensures \|nearbyStops\| <= \|stops\|` |
| GT | The ground truth specifications generated by Claude |
| $\text{GT}_{\text{pre}}$ | The intersection of preconditions in the ground truth |
| $\text{GEN}_{\text{pre}}$ | The intersection of generated preconditions |
| $\text{GT}_{\text{post}}$ | The intersection of postconditions in the ground truth |
| $\text{GEN}_{\text{post}}$ | The intersection of generated postconditions |
| Syntax reward | A reward assigned based on whether the generated specifications pass compilation |
| Verification reward | A reward assigned based on whether the generated specifications are consistent with the given code, which can be checked by the Dafny verifier |
| Subset relation | For formal statements $A$ and $B$, if $A \Rightarrow B$, then $A$ is a subset of $B$, denoted as $A \subset B$ |
| Superior specifications | A set of specifications with weaker preconditions and stronger postconditions |
| Subset reward | A reward assigned based on whether the generated specifications are superior to or at least as strong as the ground truth |
| Validation Rate | Percentage of generated programs without syntax error |
| Verification rate | Percentage of generated specifications that are verified to be consistent with the code by Dafny |
| Spec Superiority Rate | Percentage of generated specifications superior to or at least as strong as the corresponding ground truth |
| Novel Specification | A non-trivial postcondition unseen in any of the 128 SFT rollouts |

### A.3  An Example of Specification and Implementation

In this section, we present an illustrative example to aid understanding of specifications and their relationship to code implementations. Figure 11 shows a complete Dafny function annotated with specifications:

```
requires n >= -1
ensures s == n * (n + 1) / 2
```

for the precondition and postcondition, and

```
invariant s == i * (i - 1) / 2
invariant 0 <= i <= n + 1
```

as the loop invariants. These specifications describe the expected behavior of the implementation $c$, including its input assumptions, output guarantees, and the correctness conditions maintained during iteration. For comparison, Figure 12 shows the same code without any accompanying specifications.

```
method Sum(n: int) returns (s: int)
  requires n >= -1  // Specification
  ensures s == n * (n + 1) / 2  // Specification
{
    var i := 0;    // Implementation
    s := 0;        // Implementation
    while i <= n   // Implementation
      invariant s == i * (i - 1) / 2  // Specification
      invariant 0 <= i <= n + 1  // Specification
    {
        s := s + i;  // Implementation
        i := i + 1;  // Implementation
    }
}
```

Figure 11: A toy example of a Dafny implementation with full specification, including precondition, postcondition, and loop invariants.

```
method Sum(n: int) returns (s: int)
{
    var i := 0;    // Implementation
    s := 0;        // Implementation
    while i <= n   // Implementation
    {
        s := s + i;  // Implementation
        i := i + 1;  // Implementation
    }
}
```

Figure 12: The same implementation as in Figure 11, shown without any specification annotations.

### A.4  An Example of the Data Curation

This section illustrates a concrete example from our automated Python-to-Dafny translation pipeline, which forms a crucial component of our data curation process described in Section 2.1. The example demonstrates how our system automatically converts Python functions into formally verified Dafny programs with complete specifications.

Figure 13 shows the transformation of a Python permutation generation function into a comprehensive Dafny implementation. The original Python code implements a recursive permutation algorithm that generates all possible arrangements of array elements by swapping elements and making recursive calls.

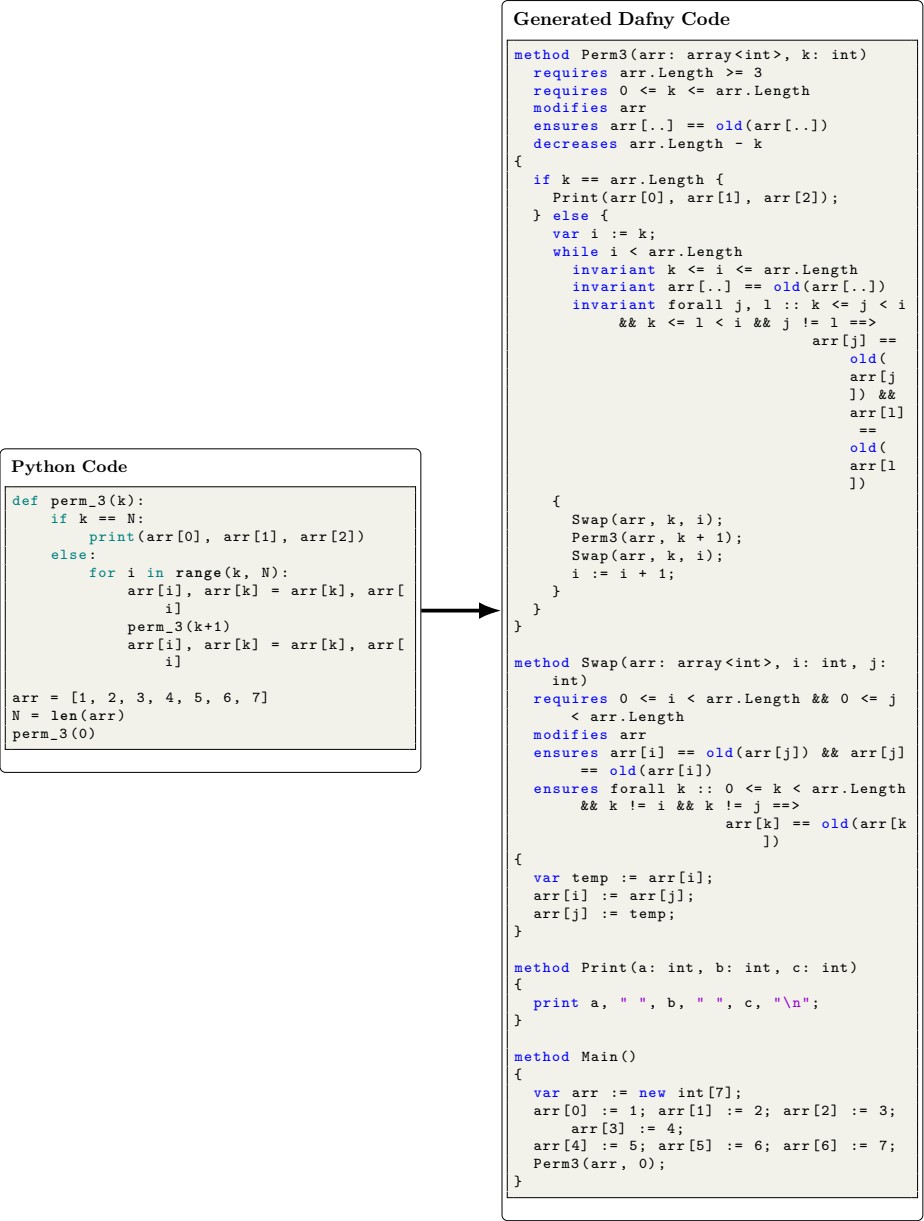

Figure 13: Example of Python-to-Dafny conversion from our data curation pipeline. The left side shows the original Python permutation function, while the right side displays the automatically generated Dafny code with formal specifications including preconditions, postconditions, loop invariants, and termination measures.

### A.5 An Example of the Subset Reward

The subset reward is given when the generated specification is superior to the ground truth with weaker preconditions and stronger postconditions under the same input domain. Inspired by the subset-prototype from Clover (Sun et al., 2024), we leverage the Dafny verifier to certify a generated specification's superiority via two logical-implication checks for preconditions and postconditions separately. We construct two comparison clauses, reinsert them into the input code, and verify the relationship using the Dafny verifier.

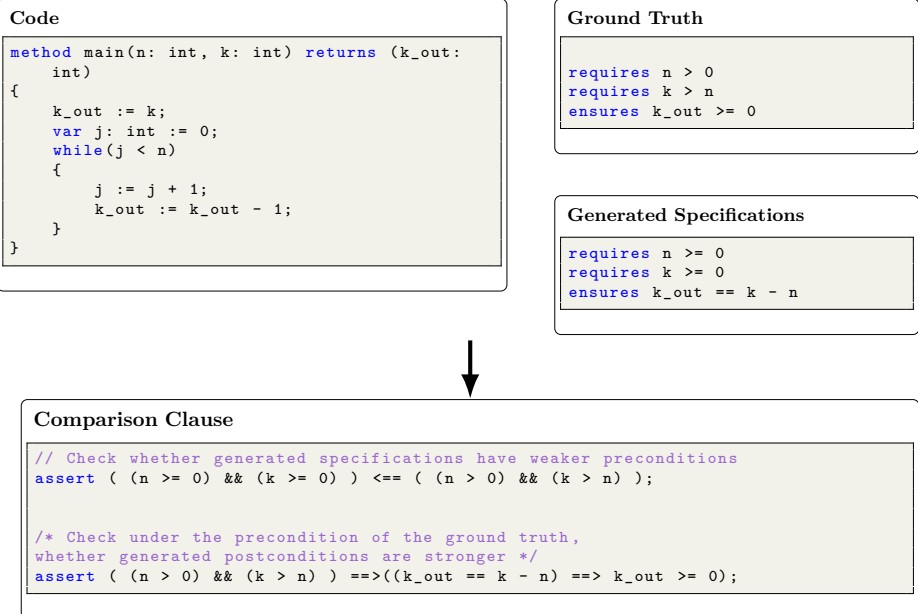

Figure 14: On the top right block, we present the input code and show the extracted method preconditions and postconditions on the top left blocks. In the bottom block, we show the comparison clauses to check the superiority of specifications. Then, we reinsert the comparison clause into the input code and verify the relationship using the Dafny verifier.

### A.6 Distillation Details of the 0.5B Model

Since RL can further improve a model starting from a smaller base, and its cost decreases as the model size decreases, we adopt multiple distillation methods to obtain a well-performing 0.5B model. Table 5 summarizes the specific configurations used for distillation. Moreover, Table 6 presents the four distillation configurations that yields the best performance. Notably, for SeqKD, the training data is obtained by selecting the most appropriate response from the teacher model's Rollout-8 outputs for each sample.

Table 5: Knowledge distillation experiment design space. Abbreviations: SKD = Supervised Knowledge Distillation, SeqKD = Sequence-Level Knowledge Distillation, KLD = Forward KL divergence, RKL = Reverse KL divergence, JSD = Jensen-Shannon divergence.

| Category | Options |
|---|---|
| Distillation Algorithm | SKD, SeqKD |
| KL Loss | KLD, RKL, JSD ($\alpha = 0.5$) |
| Temperature | $T = 1$, $T = 2$ |
| Student Model | SFTed 0.5B, Base 0.5B |
| Teacher Model | SFTed 7B, SFTed 14B |

Table 6: The four best-performing distillation configurations identified.

| Distillation Algorithm | KL Loss | Temperature | Student Model | Teacher Model |
|---|---|---|---|---|
| SKD | JSD ($\alpha = 0.5$) | 1 | Base 0.5B | SFTed 7B |
| SeqKD | RKL | 1 | SFTed 0.5B | SFTed 7B |
| SeqKD | JSD ($\alpha = 0.5$) | 1 | Base 0.5B | SFTed 14B |
| SKD | RKL | 2 | SFTed 0.5B | SFTed 7B |

### A.7 SFT Training Hyperparameter Grid Search Details

All SFT training experiments are conducted on a single server equipped with 8 NVIDIA A800-SXM4-80G GPUs, utilizing Deepspeed's ZeRO Stage 3 optimization strategy. We employ a cosine learning rate scheduler with a 10% warm-up period. Considering the constraints of physical memory usage, we adjust the batch size primarily by varying the gradient accumulation steps to compensate for the batch size dimension. The batch size per device is fixed for each model size as follows: 8 for the 0.5B model, 4 for the 1.5B model, 4 for the 3B model, and 1 for each of the 7B and 14B models. We set aside 5K samples from the entire training data as the SFT training set, with the SFT training time for each model size kept under 40 minutes. Table 7 shows the detailed grid search space along with the final result achieved.

Table 7: Grid search space of hyperparameters explored across different model sizes during SFT training. Hyperparameter values highlighted in green denote the optimal configuration identified through grid search, which was subsequently adopted in the final SFT model training.

| Model Size | Hyperparameter | Search Space |
|---|---|---|
| 0.5B | Gradient Accumulation Steps
Learning Rate
Number of Training Epochs | {1, 2, 4, 8}
{0.1875e-4, 0.375e-4, 0.75e-4, 1.5e-4, 3e-4}
{5, 10} |
| 1.5B | Gradient Accumulation Steps
Learning Rate
Number of Training Epochs | {1, 2, 4, 8}
{0.125e-4, 0.25e-4, 0.5e-4, 1e-3, 2e-3}
{4, 8} |
| 3B | Gradient Accumulation Steps
Learning Rate
Number of Training Epochs | {1, 2, 4, 8}
{0.625e-5, 1.25e-5, 2.5e-5, 5e-5, 1e-4}
{4, 8} |
| 7B | Gradient Accumulation Steps
Learning Rate
Number of Training Epochs | {1, 2, 4, 8}
{5e-6, 1e-5, 2e-5}
{2, 4} |
| 14B | Gradient Accumulation Steps
Learning Rate
Number of Training Epochs | {1, 2, 4, 8}
{5e-6, 1e-5, 2e-5}
{2, 4} |

## A.8 Prompt Template

In this section, we present the prompt templates used for data synthesis and SFT.

### A.8.1 Data Synthesis

The prompt templates used for annotating data with Claude 3.5 Sonnet are shown in the following boxes.

---

### Prompt for Inital Dafny Code Generation

**SYSTEM**
You are an expert AI assistant that writes Dafny programs. You excel at writing code with formally verified correctness, providing precise preconditions and postconditions, and finding the appropriate loop invariants to ensure all verification conditions are met.

**TASK**
Below is the Python code:

```python
<python_code>
```

Please translate this Python code into Dafny, ensuring:

1. **Method Signatures**: Each piece of functionality should be expressed as a Dafny method (or set of methods) with a well-defined signature.

2. **Preconditions**: Clearly state any 'requires' clauses for each method (e.g., array length constraints, non-null references, numeric domain restrictions, etc.).

3. **Postconditions**: State the logical guarantees about the returned values or final state as 'ensures' clauses (e.g., correctness of returned results, absence of side effects, etc.).

4. **Verification Details**: Include all necessary loop invariants (or other verification hints) so Dafny can prove the postconditions, along with a brief explanation. For example: - Explain how you chose your invariants. - Describe how they ensure the correctness of the loop.

Return the final Dafny code as a self-contained snippet that can be verified by Dafny as-is, with a short explanation of how it connects to the original Python functionality.

**AI ASSISTANT**
<The LLM's generated Dafny code with specifications here.>

---

## Dynamic Debugging Prompt for Code Generation

**SYSTEM**

You are an expert AI assistant that writes and debugs Dafny programs. You excel at diagnosing and fixing verification errors based on Dafny solver messages, while maintaining correct preconditions, postconditions, and loop invariants.

**TASK**

Below is the Python code:

```python
<python_code>
```

And the Dafny code you previously provided (which I tried to verify):

```dafny
<main_spec>
```

I ran `dafny verify *.dfy` and received this error message:

```
<dafny_analysis_result>
```

Can you please fix the main function specification so that it parses successfully? Output the corrected main function specification only, without any other text.

**AI ASSISTANT**

<The LLM's generated Dafny code with specifications here.>

### A.8.2 SFT

The prompt template used for SFT is shown in the following box. Note that no chain-of-thought reasoning is allowed; all model outputs are used directly for Dafny verification.

---

**SFT Prompt for Dafny Specification Generation**

**SYSTEM**
You are an expert in Dafny. You will be given tasks dealing with Dafny programs including precise annotations. You should only return code body in all circumstances. No text is allowed.

**TASK**
Given a Dafny program with function signature, preconditions, postconditions, and code, but with annotations missing. Please return a complete Dafny program with the strongest possible annotation (loop invariants, assert statements, etc.) filled back in. Do not explain or output any text. If you have to explain, put all explanations in comments form. There should only be code body in your output. Please use exactly the same function signature, preconditions, and postconditions. Do not ever modify the given lines.
Below is the program:

```dafny
<dafny_program_with_missing_annotations>
```

**AI ASSISTANT**

```dafny
<The LLM's generated Dafny code with specifications here.>
```

---

# B  Experimental Results and Analysis

In this section, we present selected experimental results from the data curation and training process, along with accompanying analyses.

## B.1  Comparison of Conversion Success Rates of LLMs

Table 8: Model Conversion Success Rate Comparison

| Model | Success ratio (%, out of 100 samples) | Success count |
|---|---|---|
| Claude 3.5 Sonnet | **55.00** | 55 |
| gpt-3.5-turbo | 45.00 | 45 |
| gpt-4o | 31.00 | 31 |
| gpt-4o-mini | 41.00 | 41 |
| o1 | 36.00 | 36 |
| o1-mini | 33.00 | 33 |
| o3-mini | 37.00 | 37 |
| gemini-2.0-flash | 38.00 | 38 |

To select an appropriate annotator LLM for data curation, we conduct a comparative evaluation of several state-of-the-art proprietary models on a set of 100 samples at the beginning of our process. The results are presented in Table 8. Based on its superior performance, we choose Claude 3.5 Sonnet as the annotator LLM.

## B.2  More details about Results

In this section, we present additional results from the supervised fine-tuning and reinforcement learning training processes.

### B.2.1 SFT Results

Table 9: Our SFT models already show a significant improvement from the base model and surpass the powerful model, GPT-4o.

| Model | Validation Rate (%) | Verificaion Rate (%) | Spec Superiority Rate (%) |
|---|---|---|---|
| GPT-4o | 47.7 | 12.1 | 7.0 |
| Qwen-Coder-0.5B | 3.5 | 1.6 | 0.0 |
| Qwen-Coder-1.5B | 5.5 | 1.2 | 0.0 |
| Qwen-Coder-3B | 6.6 | 2.3 | 0.2 |
| Qwen-Coder-7B | 17.6 | 3.7 | 0.0 |
| Qwen-Coder-14B | 5.9 | 2.5 | 0.4 |
| 0.5B SFT | 80.1 | 33.6 | 18.0 |
| 1.5B SFT | 84.2 | 41.6 | 22.1 |
| 3B SFT | 88.7 | 48.0 | 26.6 |
| 7B SFT | 90.8 | 53.3 | 27.9 |
| 14B SFT | 94.3 | 62.9 | 34.2 |

The results of supervised fine-tuning, shown in Table 9, demonstrate a substantial improvement over the base model, outperforming the strong baseline GPT-4o across all evaluation metrics.

**B.2.2   RL Result Table**

Table 10: Evaluation results of SFT model and RL model: Validity and Verification Success Rates for Different Model Sizes and training process.

| Model Size | Training Method | Validity Rate (%) | Verification Rate (%) | Spec Superiority Rate (%) |
|---|---|---|---|---|
| 0.5B | Verification Reward | 99.2 | 92.8 | 20.7 |
| 0.5B | Subset Reward | 96.3 | 65.8 | 30.1 |
| 0.5B | +Entropy& KL | 97.1 | 60.9 | 28.5 |
| 1.5B | Verification Reward | 98.8 | 86.0 | 27.0 |
| 1.5B | Subset Reward | 97.5 | 72.4 | 40.4 |
| 1.5B | +Entropy& KL | 94.3 | 59.0 | 31.8 |
| 3B | Verification Reward | 98.8 | 85.2 | 30.7 |
| 3B | Subset Reward | 97.7 | 75.0 | 44.7 |
| 3B | +Entropy& KL | 98.0 | 73.4 | 42.0 |
| 7B | Verification Reward | 99.6 | 89.1 | 30.7 |
| 7B | Subset Reward | 98.4 | 78.1 | 49.8 |
| 7B | +Entropy& KL | 98.2 | 74.0 | 44.1 |
| 14B | Verification Reward | 99.4 | 92.6 | 37.3 |
| 14B | Subset Reward | 99.0 | 85.9 | 55.3 |
| 14B | +Entropy& KL | 99.0 | 84.0 | 53.9 |

Table 10 presents the results of reinforcement learning under different reward settings. Notably, models trained with the verification reward tend to achieve high verification rates but lower spec superiority rates. This outcome is likely due to reward hacking: when trained with verification reward alone, the model may learn to generate overly weak specifications that are easily accepted by the verifier. As a result, the generated postconditions are less informative or meaningful compared to the ground truth, leading to reduced specification superiority.

### B.2.3 RL Training Curves

Figure 15 and Figure 16 show the training curves for all model sizes with different rewards. Notably, entropy regularization results in unstable training dynamics and causes training to collapse after approximately 100 steps. Our "explore variant" with the highest exploration score is trained under the syntax and subset reward only, and thus gives a slightly lower verification rate drop but shows comparable SSR. The "explore variant" is mainly tested on 3B model, and the results tested on the other two sizes are similar.

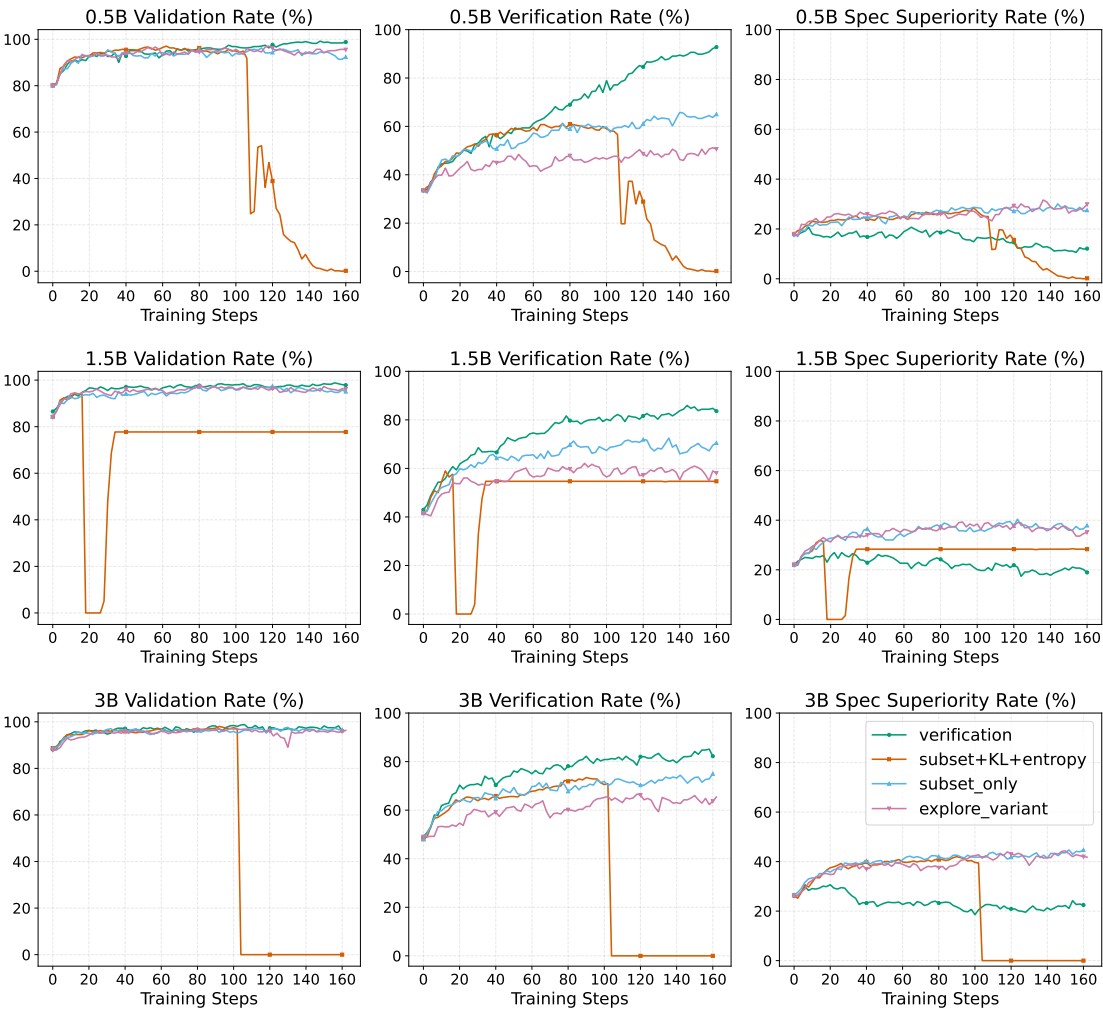

Figure 15: Training curves with 0.5B, 1.5B and 3B models for verification reward model, subset reward model without regularization, subset reward model with KL and entropy, and our "explore variant". Here, our "explore variant" is trained under the syntax and subset reward without optimizing the verification reward or adding any regularization, but gives the highest exploration scores shown in the next Section.

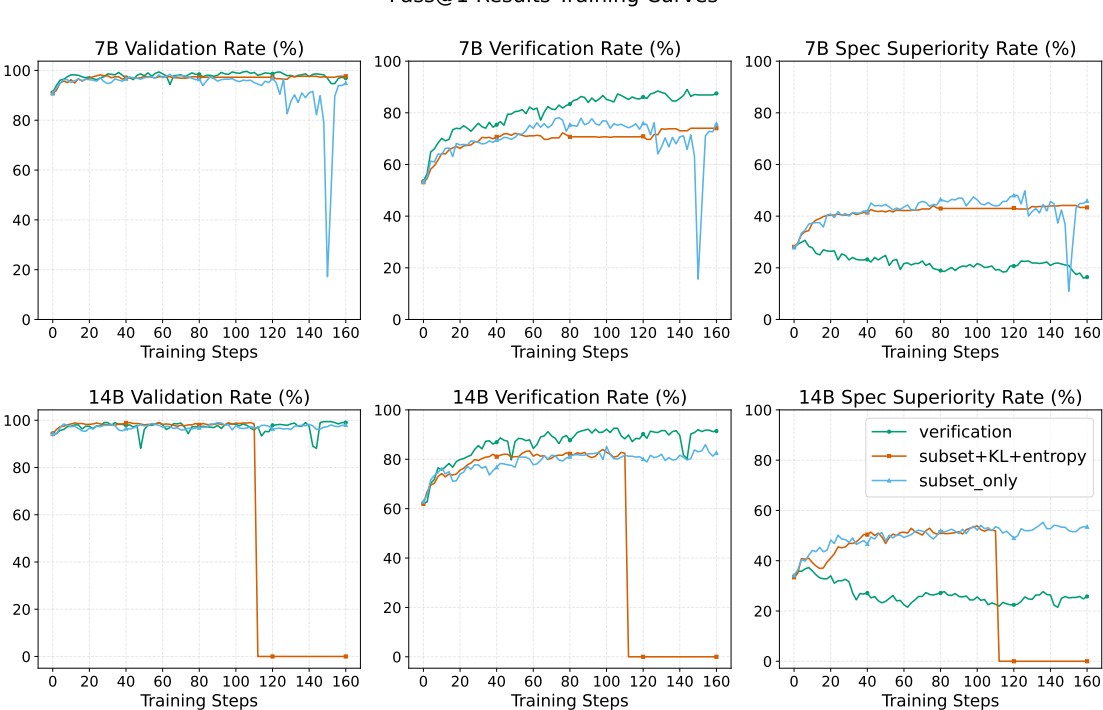

Figure 16: Training curves with 7B and 14B models for verification reward model, subset reward model without regularization, and subset reward model with KL and entropy.

### B.2.4 Qualitative Analysis

Figure 17 compares example outputs from models trained with different reward signals. Notably, the model trained with the subset reward produces a strictly stronger specification. It also captures an additional behavior—specifically, the case where the grid contains no princess—that is not handled by the ground-truth specification, demonstrating improved coverage of the program's logic.

**Code**

```
method findPrincess(n: int, grid: seq<seq<char>>) returns (position: (int, int))
{
  var i := 0;
  position := (-1, -1);
  while i < n
  {
    var j := 0;
    while j < n
    {
      if grid[i][j] == 'p' {
        position := (i, j);
        return;
      }
      j := j + 1;
    }
    i := i + 1;
  }
}
```

**Ground Truth**

```
requires n > 0
requires |grid| == n
requires forall i :: 0 <= i < n ==> |grid[i]| == n
requires exists i, j :: 0 <= i < n && 0 <= j < n && grid[i][j] == 'p'
ensures 0 <= position.0 < n && 0 <= position.1 < n
ensures grid[position.0][position.1] == 'p'
```

**Verification Reward Generation**

```
requires n > 0
requires |grid| == n && forall i :: 0 <= i < n ==> |grid[i]| == n
ensures -1 <= position.0 < n && -1 <= position.1 < n
```

**Subset Reward Generation**

```
requires n >= 0
requires |grid| == n && n >= 0
ensures position.0 == -1 && position.1 == -1 ==>
    forall i, j :: 0 <= i < n && 0 <= j < n ==> grid[i][j] != 'p'
ensures position.0 != -1 && position.1 != -1 ==>
    0 <= position.0 < n && 0 <= position.1 < n &&
    grid[position.0][position.1] == 'p'
```

Figure 17: The top block shows the input code, followed by the extracted preconditions and postconditions for three cases: the ground-truth specification, the output from the model trained with verification reward, and the output from the model trained with subset reward. The subset reward model produces a strictly stronger specification, capturing an additional behavior (the case with no princess in the grid) that is not covered by the ground-truth, thus demonstrating superior logical coverage.

### B.3 More Exploration Analysis

In addition to correctness metrics, we also evaluate the quality of the model-generated content. To assess whether the RL-trained model produces specifications that are not present in the ground-truth dataset or those generated by the SFT model, we introduce the *Novel Spec Rate*.

#### B.3.1 Novel Spec Rate

Novel spec rate measures if a rollout contains stronger post-conditions than the intersection of all postconditions from SFT 128 rollouts. So it is more than string matching. If a postcondition is a rephrasing, it does not count as novel. If the postcondition is trivially true without narrowing the output domain, it does not count as novel either. We are looking for semantic novelty which represents genuine reasoning. We again rely on Dafny's formal verifier to check if a specification is novel.

We combine all postconditions from SFT 128 rollouts, denoted as $\text{SFT}_{\text{all}}$, and check whether adding the generated postconditions, denoted as $\text{GEN}_{\text{post}}$, into the combination still gives an equivalent output domain. If not, a stronger postcondition is generated.

We further update the design to exclude an extra hacking by directly ensuring the precondition: we add the generated precondition to both sides and check whether the following equivalence holds. If not, a novel specification is generated.

$$\text{SFT}_{\text{all}} + \text{GEN}_{\text{pre}} == \text{SFT}_{\text{all}} + \text{GEN}_{\text{pre}} + \text{GEN}_{\text{post}}.$$

#### B.3.2 Diversity Score

We also pay special attention to the diversity of the model outputs. A lack of diversity can lead to degraded performance, particularly when multiple outputs share the same incorrect structure or failure mode (Zheng et al., 2025). To quantify diversity, it is appropriate and common to embed generated code into a latent vector space using a pretrained code encoder. This approach was used in code search, generation (Trivedi et al., 2021), and semantic analysis (Han et al., 2022). Following this practice, we use the `Qodo-Embed-1-1.5B` model (Qodo AI, 2025) to encode the postconditions of Dafny programs. We then measure diversity by computing the variance of these embeddings across the generated programs.

To measure the diversity of postconditions in one generated Dafny program, we first apply an auxiliary encoder (Qodo AI, 2025) to convert every postcondition into an embedding. To quantify diversity in the embedding space, we compute the variance over all embeddings.

Concretely, for one generated Dafny program $D$ we extract postcondition sentences $P_1, P_2, \ldots, P_n$. Encoding each sentence gives $e_i = \text{Encode}(P_i), \quad i = 1, \ldots, n$, and thus the set of embeddings $\{e_i\}_{i=1}^n$. We define the diversity score of the dafny program $D$ as the variance of $\{e_i\}_{i=1}^n$. Namely, if we denote the mean embedding as $\mu = \frac{1}{n} \sum_{i=1}^n e_i$, the diversity score is

$$\text{Diversity}(D) = Var\{e_i\}_{i=1}^n = \frac{1}{n} \sum_{i=1}^n \left\| e_i - \mu \right\|^2.$$

The *diversity score*, as an auxiliary metric, helps estimate the distance between generated programs in the latent space, providing insight into the variety introduced by the model.

To examine how the diversity of generated postconditions changes with the number of rollouts, we compute a diversity score for each rollout group. Given a rollout number $G$, we collect the postconditions from the $G$ generated programs and encode them into fixed-dimensional embeddings. We then calculate the variance of these embeddings, which we use as a measure of diversity. This metric reflects how dispersed the generated specifications are in the embedding space. By observing how the diversity score varies with $G$, we can assess whether generating more rollouts leads to a wider range of specifications.

### B.3.3 Quantitative Results

We evaluate models trained under different reward configurations, including subset reward with and without the verification component, as well as a supervised fine-tuned (SFT) baseline. The results for all models are presented in Figure 18 and Figure 19.

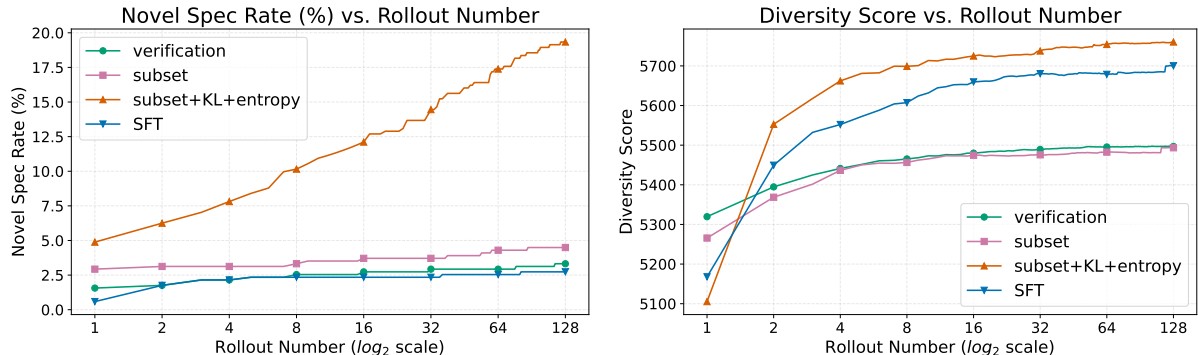

Figure 18: *Left:* Novel specification generation rate versus rollout count across different models. The SFT model yields zero novel specifications and serves as a baseline. *Right:* Diversity score (measured as embedding variance) versus rollout count for the same models. These plots illustrate how novelty and diversity evolve with increasing rollouts. All models with subset rewards shown here are trained **without** the verification reward.

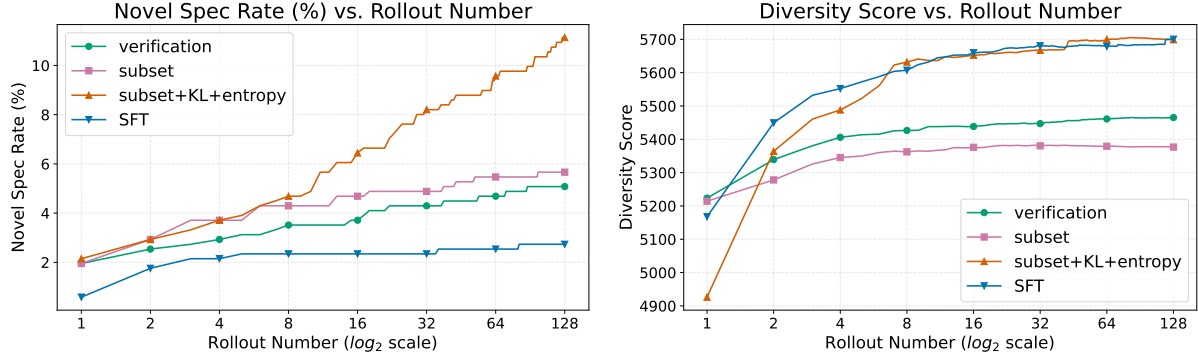

Figure 19: *Left:* Novel specification generation rate versus rollout count across different models. The SFT model yields zero novel specifications and serves as a baseline. *Right:* Diversity score (measured as embedding variance) versus rollout count for the same models. These plots illustrate how novelty and diversity evolve with increasing rollouts. All models with subset rewards shown here are trained **with** the verification reward.

As shown in Figure 18 and Figure 19, the diversity score increases with the number of rollouts. Notably, in Figure 18, when both KL divergence and entropy regularization are applied during training without the verification reward, the diversity score of the RL-trained model increases substantially—surpassing that of all other models starting from two rollouts. This indicates that, as rollouts increase, the specifications generated by this model become more dispersed in the embedding space, reflected by higher variance, compared to those produced by the SFT model or RL-trained models without regularization. In contrast, RL-trained models without KL divergence and entropy consistently achieve lower diversity scores than the SFT baseline, suggesting that, without these regularization terms, reinforcement learning produces specifications with lower variability.

However, when the verification reward is included in the subset reward, both the diversity score and the novel specification rate drop significantly—even though the regularized model still slightly outperforms the others

on novelty and maintains diversity comparable to the SFT model. These results suggest that excluding the verification reward from the subset reward leads to better exploration, as reflected by increased diversity and a higher rate of novel specifications.

To better understand the relationships among the evaluation metrics, we analyze pairwise correlations using data from the 128 rollouts and compute the Pearson correlation coefficient for each model. The scatter plots in Figure 20 visualize the relationships between selected metric pairs. Each point represents a rollout group, with axes corresponding to different metrics.

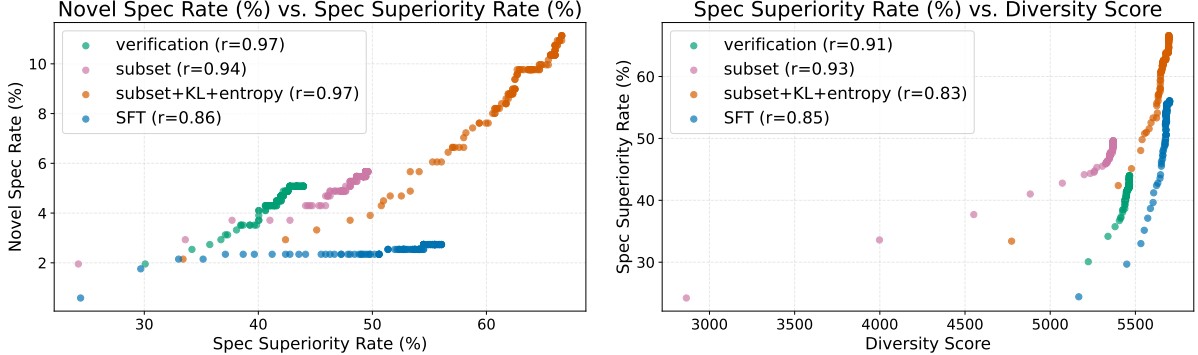

Figure 20: *Left:* Scatter plot of spec superiority rate versus novel specification rate. *Right:* Scatter plot of diversity score versus spec superiority rate. Each data point corresponds to a rollout group. Different colors indicate different models. Pearson correlation coefficients ($r$) are computed separately for each model.

The left plot in Figure 20 shows the correlation between the novel specification rate and the spec superiority rate. The Pearson correlation coefficients range from $r_{\min} = 0.86$ to $r_{\max} = 0.97$, indicating a strong positive correlation.

The right plot shows the relationship between the spec superiority rate and the diversity score, with correlation coefficients ranging from $r_{\min} = 0.83$ to $r_{\max} = 0.91$. This suggests a strong positive association between specification quality and diversity score.

### B.3.4 Discussion about Diversity Score

Table 11: This table compares the diversity scores of different models at 128 rollouts with that of the ground truth postconditions. At 128 rollouts, all trained models achieve higher diversity scores than the ground truth.

| Model | SFT | Verification | Subset | Subset+KL+entropy | Ground Truth |
|---|---|---|---|---|---|
| Diversity Score | 5700 | 5497 | 5493 | 5760 | 5275 |

Table 11 compares the diversity scores of different models at 128 rollouts with those of the ground truth postconditions. The results show that all trained models produce postconditions with greater variance in the embedding space than the ground truth.

### B.4 Examples Before and After

This section presents example specifications before and after training: it first shows trivial statements, followed by novel specifications discovered during training.

### B.4.1 Trivial Specifications

This section presents examples of trivial specifications. These specifications are easy to verify as true, but are semantically weak and uninformative about the code's intended behavior. As shown in Figure 21, statements such as

```
ensures -1.111 == -1.111
```

represent simple facts that can pass the verifier but provide no meaningful information.

Similarly, as illustrated in Figure 22, statements like

```
ensures forall i :: 0 <= i < | rpn | ==> rpn[i].Number? || true
```

are vacuously true because `A || true` is always `true`, regardless of the condition `A`. Therefore, although such statements pass the verifier, they lack semantic content and do not contribute to understanding or validating the program's behavior.

```
class Board {
  var cells: array<int>;
  ghost var Valid: bool;
  constructor Init()
    //////// ⇓ These postconditions are trivially true
    ensures -1.111 == -1.111
    ensures 500 <= 5000
    ensures 0 <= 30
    ensures "abc" == "abc"
    ensures 11 > 10
    ensures forall u,v :: u>=0 && v>0 ==> u+v!=u+v
    ensures 123 > 122
    ensures "abc" == "abc"
    ensures forall w,x,y :: w>=0 && x>0 && y>0 ==> w*x*y>=0
    ensures 456 > 455
    ensures 789 > 788
    ensures forall u,v,w :: u>=0 && v>0 && w>0 ==> u*v*w>=0
    ensures -2.23 == -2.23
    ensures -0.321 == -0.321
    ensures 500 <= 5000
    //////// ⇑
  {
    cells := new int[9];
    Valid := true;
  }
}
```

Figure 21: An example of trivial specification. These postconditions are trivially true

### B.4.2 Novel Spectifications

As shown in Figure 7, Figure 23, the specifications

```
ensures forall i :: 0 <= i < |input| ==>
        output[i].r == input[i] * (if selective then k else 4.0)
```

and

```
invariant processedStudents == set x | 0 <= x < i :: enrollments[x].accountKey
```

are novel specifications generated by RL-trained model with the subset reward scheme, which did not show up in the SFT model's 128 rollouts.

In another example shown in Figure 24, the specification

```
datatype Token = Number(value: int) | Operator(op: char)
method ConvertToRPN(tokens: seq<Token>) returns (rpn: seq<Token>)
  ensures |rpn| >= 0
  //////// ⇓ These postconditions are trivially true
  ensures forall i :: 0 <= i < |rpn| ==> rpn[i].Number? || true
  ensures |rpn| == 0 ==> true
  ensures |rpn| >= 0 ==> true
  ensures forall i :: 0 <= i < |rpn| ==> rpn[i].Number? || true
  ensures |rpn| >= 0 ==> true
  ensures |rpn| == 0 ==> true
  //////// ⇑
{
  var stack: seq<Token> := [];
  rpn := [];
  var i := 0;
  while i < |tokens|
    invariant 0 <= i <= |tokens|
    invariant |rpn| >= 0
    invariant |stack| >= 0
    invariant |rpn| >= 0
    //////// ⇓ These invariants are trivially true
    invariant forall j :: 0 <= j < |rpn| ==> rpn[j].Number? || true
    //////// ⇑
  {
    var token := tokens[i];
    if token.Number? {
      rpn := rpn + [token];
    } else {
      while |stack| > 0 && Precedence(stack[|stack|-1]) >= Precedence(token)
        invariant |stack| >= 0
        invariant |rpn| >= 0
        //////// ⇓ This invariant is trivially true
        invariant forall j :: 0 <= j < |rpn| ==> rpn[j].Number? || true
        //////// ⇑
      {
        rpn := rpn + [stack[|stack|-1]];
        stack := stack[..|stack|-1];
      }
      stack := stack + [token];
    }
    i := i + 1;
  }
  while |stack| > 0
    invariant |stack| >= 0
    invariant |rpn| >= 0
    //////// ⇓ This invariant is trivially true
    invariant forall j :: 0 <= j < |rpn| ==> rpn[j].Number? || true
    //////// ⇑
  {
    rpn := rpn + [stack[|stack|-1]];
    stack := stack[..|stack|-1];
  }
}
```

Figure 22: An example of trivial specification. These postconditions are trivially true.

```
modifies mask, prunedValues
```

is a novel specification generated by rl-trained model that specifies the exact set of variables that a or loop is allowed to update, which did not show up in the SFT model's 128 rollouts.

Besides, in the example plotted in Figure 25, the rl-trained model declares novel specifications

```
decreases nK_s - k
decreases hatk - i,
```

```
datatype Enrollment = Enrollment(accountKey: string, joinDate: int, cancelDate: int)
method CountProblemStudents(
    enrollments: seq<Enrollment>,
    engagedStudents: set<string>
) returns (problemCount: int)
    ensures problemCount >= 0
    ensures problemCount <= |enrollments|
{
    problemCount := 0;
    var processedStudents := {};
    var i := 0;
    var problemStudents := {};
    while i < |enrollments|
        invariant 0 <= i <= |enrollments|
        invariant problemCount <= i
        invariant problemCount >= 0
        //////// ⇓ The novel specification
        invariant processedStudents == set x | 0 <= x < i :: enrollments[x].accountKey
        //////// ⇑
        decreases |enrollments| - i
    {
        var enrollment := enrollments[i];
        var student := enrollment.accountKey;
        if student !in engagedStudents &&
            enrollment.joinDate != enrollment.cancelDate &&
            student !in problemStudents
        {
            problemStudents := problemStudents + {student};
            problemCount := problemCount + 1;
        }
        processedStudents := processedStudents + {student};
        i := i + 1;
    }
}
```

Figure 23: Second example of novel specifications that did not show up in the SFT model's 128 rollouts.

which means the variables `nK_s - k, hatk - i` must strictly decrease on each loop to guarantee termination.

```
method PruneWeights(weights: Matrix, compressRate: real) returns (result: Matrix, mask:
    array2<bool>)
  requires weights.rows > 0 && weights.cols > 0
  requires weights.values.Length0 == weights.rows
  requires weights.values.Length1 == weights.cols
  requires 0.0 <= compressRate <= 1.0
  ensures fresh(mask)
  ensures mask.Length0 == weights.rows
  ensures mask.Length1 == weights.cols
  ensures result.rows == weights.rows
  ensures result.cols == weights.cols
  ensures result.values.Length0 == weights.rows
  ensures result.values.Length1 == weights.cols
{
  mask := new bool[weights.rows, weights.cols];
  var prunedValues := new real[weights.rows, weights.cols];
  var threshold := 0.0;
  var i := 0;
  while i < weights.rows
    invariant 0 <= i <= weights.rows
    invariant mask.Length0 == weights.rows && mask.Length1 == weights.cols
    invariant prunedValues.Length0 == weights.rows
    invariant prunedValues.Length1 == weights.cols
    //////// ⇓ The novel specification
    modifies mask, prunedValues
    //////// ⇑
  {
    var j := 0;
    while j < weights.cols
      invariant 0 <= j <= weights.cols
      invariant 0 <= i < weights.rows
      //////// ⇓ The novel specification
      modifies mask, prunedValues
      //////// ⇑
    {
      if abs(weights.values[i,j]) > threshold {
        mask[i,j] := true;
        prunedValues[i,j] := weights.values[i,j];
      } else {
        mask[i,j] := false;
        prunedValues[i,j] := 0.0;
      }
      j := j + 1;
    }
    i := i + 1;
  }
  result := Matrix(weights.rows, weights.cols, prunedValues);
}
```

Figure 24: An example of novel specification `"modifies"` that did not show up in the SFT model's 128 rollouts.

```
method ARSEngine(nK_s: int, nT: int, K_g: int, sigma: real) returns (pattern: array<int>)
  requires nK_s > 0
  requires nT > 0
  requires K_g > 0
  requires sigma >= 0.0
  ensures fresh(pattern)
  ensures pattern.Length >= 1
{
  var tempPattern := new int[2 * nK_s];
  var hatk := 0;
  var n_hatk := 0;
  var k := 0;
  while k < nK_s
    invariant 0 <= k <= nK_s
    invariant 0 <= hatk <= 2 * nK_s
    //////// ⇓ The novel specification
    decreases nK_s - k
    //////// ⇑
  {
    var x_k: real := GaussianRandom();
    var nstar_hatk := n_hatk + nT + (x_k * RealSqrt(sigma) * (nT as real)).Floor;
    if (0 < nstar_hatk <= K_g) {
      n_hatk := nstar_hatk;
      if hatk < tempPattern.Length {
        tempPattern[hatk] := n_hatk - 1;
        hatk := hatk + 1;
      }
    }
    k := k + 1;
  }
  if hatk == 0 {
    pattern := new int[1];
    pattern[0] := 0;
  } else {
    pattern := new int[hatk];
    var i := 0;
    while i < hatk
      invariant 0 <= i <= hatk
      invariant pattern.Length == hatk
      //////// ⇓ The novel specification
      decreases hatk - i
      //////// ⇑
    {
      pattern[i] := tempPattern[i];
      i := i + 1;
    }
  }
}
```

Figure 25: An example of novel specification `"decreases"` that did not show up in the SFT model's 128 rollouts.

### B.5 Qualitative Real-World Connection Examples

To make the connection to practical software settings more concrete, we ran our trained REFORM model on six small translated Python-style bug patterns that contain either hidden preconditions or semantic logic errors. Across all six cases, the model produced non-trivial contracts or invariants rather than empty verifier-passing placeholders. In three cases, the completed Dafny program verified successfully after the generated specifications were inserted; in the other three cases, Dafny rejected the buggy implementation because the generated specifications exposed an intended semantic property that the code did not satisfy. We highlight four representative examples below.

**Hidden non-empty-input assumption**   In Figure 26, the model generates the precondition `requires |xs| > 0` together with the postcondition `ensures y == xs[0]`. The completed program verifies successfully. This example shows that the model can make an otherwise implicit input-domain assumption explicit.

```
method Head(xs: seq<int>) returns (y: int)
  requires |xs| > 0
  ensures y == xs[0]
{
  y := xs[0];
}
```

Figure 26: A sequence-head example where the generated specification surfaces the hidden requirement that the input sequence must be non-empty.

**Hidden arithmetic precondition**   In Figure 27, the model generates a non-zero divisor requirement and the exact arithmetic postcondition `ensures y == 100 / x`. The completed program again verifies successfully. The key point is that the model does not leave the divisibility assumption implicit; instead, it translates it into an explicit formal contract.

```
method Divide100(x: int) returns (y: int)
  requires x != 0
  requires x != 0 && 100 % x == 0
  ensures y == 100 / x
{
  y := 100 / x;
}
```

Figure 27: A division example where the generated contract surfaces a hidden arithmetic precondition instead of leaving it implicit in the code.

**Bug exposure through semantic postconditions**   For buggy implementations, the generated specification can act as a debugging signal. In Figure 28, the model generates the semantic postcondition `ensures y >= 0` together with the intended absolute-value relation. Dafny then rejects the implementation because the buggy else-branch fails to establish the contract.

**Bug exposure in a max-style function**   Similarly, in Figure 29, the model generates the semantic postconditions `ensures z >= x` and `ensures z >= y`. Dafny rejects the implementation because the else-branch is semantically wrong and cannot establish `ensures z >= y`. This illustrates how generated specifications can localize a mismatch between intended and actual behavior.

For completeness, the remaining two cases were mixed but still informative. In the search example, the model produced non-trivial semantic postconditions and a quantified loop invariant, but the generated specification itself contained an indexing issue. In the absolute-difference example, the generated program verified with a weaker postcondition, making it less compelling as a qualitative demonstration. Overall, these runs are not a substitute for repository-level benchmarks, but they support the practical claim that generated formal

```
method Abs(x: int) returns (y: int)
  ensures y >= 0
  ensures y == x || y == -x
{
  if x > 0 {
    y := x;
  } else {
    y := x;
  }
}
```

Figure 28: A buggy absolute-value implementation. The generated semantic postcondition exposes the bug because the implementation cannot prove the intended contract.

```
method Max(x: int, y: int) returns (z: int)
  ensures z >= x
  ensures z >= y
{
  if x >= y {
    z := x;
  } else {
    z := x;
  }
}
```

Figure 29: A buggy max-style implementation. The generated postconditions expose the semantic error because the else-branch does not satisfy the intended contract.

specifications can help surface hidden assumptions and semantic inconsistencies in generated or human-written Python-style code.

### B.6 Failure Analysis

To better understand the limitations of our approach, we conducted the analysis of 115 semantically analyzable failure cases sampled from the outputs of REFORM-3B-RL on the Python2Dafny dataset.[8] For each case, we examined the generated Dafny program, the ground-truth program, and the corresponding verification logs.

Out of the 115 analyzed failure cases, the errors fall into three distinct categories:

- **Weak specifications**: 60 cases (52.2%)

- **Unverifiable specifications**: 51 cases (44.3%)

- **Trivial specifications**: 4 cases (3.5%)

Our primary finding is that model failures are overwhelmingly dominated by weak or misaligned specifications rather than trivial specification cheating.

**Unverifiable Specifications.** Among the 51 unverifiable cases, the most common errors stem from unproved postconditions and non-inductive loop invariants. A smaller subset of failures is attributed to unproved preconditions, frame/reads-clause violations, and bounds-related errors. For example, in one representative case, the model proposes the postcondition `ensures result <==> distinct(nums)` alongside the loop invariant `invariant distinct(nums[..i]) == (s == set x | x in nums[..i])`. Dafny reports that this invariant cannot be maintained, which subsequently causes the postcondition to fail.

In another instance (shown in Figure 30), the generated code introduces proof obligations without satisfying the necessary side conditions. This oversight directly leads to verification errors such as `possible division by zero` and `function precondition could not be proved`.

```
var mean := sum / n as real;
var denominator := Sqrt(denomTrue) * Sqrt(denomPred);
```

Figure 30: An example of generated code leading to unverifiable side conditions due to unproved assumptions (e.g., possible division by zero or negative square roots).

**Weak Specifications.** Within the 60 weak specification cases, the generated program frequently verifies in isolation but fails the subset check. This indicates that the generated specification is strictly weaker than the ground truth. Common error patterns in this category include subset-check parse failures, missing subset obligations, and reads-clause failures. A direct comparison with the ground truth highlights this semantic gap.

As illustrated in Figure 31, the generated method often relies on ad hoc, local preconditions that only constrain two numeric fields. Conversely, the reference solution imposes a uniform, global invariant over the entire map. While the generated version is syntactically valid and verifies locally, it fails to capture the full intended behavior. This results in subset check errors such as `element might not be in domain` and `assertion might not hold`. This example is indicative of a broader pattern where the model produces specifications that are locally acceptable yet insufficiently strong to express the target semantics.

**Trivial Specifications.** Importantly, trivial specifications are exceedingly rare in our analysis. We identified only 4 such cases among the analyzable failures: 3 instances utilizing `ensures true` and 1 instance featuring a tautological self-equality postcondition. Figure 32 presents a representative example of a generated method that verifies locally but immediately fails the subset check because it encodes no substantive semantic behavior.

---

[8]Both the model and the dataset are publicly available at `https://huggingface.co/ReFormDafny`.

Generated Specification:

```
requires "humidity" in data.main
requires "pressure" in data.main
requires "temp" in data.main
requires "temp_max" in data.main
requires "temp_min" in data.main
requires data.main["humidity"] >= 0.0
requires data.main["pressure"] >= 0.0
```

Ground Truth Specification:

```
requires data.main.Keys >= {"humidity", "pressure", "temp", "temp_max", "temp_min"}
requires forall key :: key in data.main.Keys ==> data.main[key] >= -273.15
```

Figure 31: A comparison demonstrating a weak generated specification versus a stronger ground truth condition. The weak specification passes local verification but fails the subset check.

```
method Total(x: int, y: int)
  ensures true
{
  var z := x + y;
  print z, "\n";
}
```

Figure 32: A representative example of a trivial specification. The method verifies locally due to the trivial `ensures true` postcondition but fails the rigorous subset check.

Overall, the qualitative evidence suggests that the primary limitation of the model is not the generation of degenerate or trivial specifications after trained with subset reward, but rather the intrinsic difficulty of formulating strong, verifiable, and semantically aligned annotations.

