# OpenReview forum: "Re:Form --- Reducing Human Priors in Scalable Formal Software Verification with RL in LLMs: A Preliminary Study on Dafny"
_TMLR — Accepted by TMLR_

### Review · Reviewer_v3Qt · 2025-12-02

**Summary Of Contributions:**

This paper introduces a pipeline for training LLMs to generate specifications for and in Dafny, a formally verifiable programming language. The authors are able to eliminate the need for human feedback by making use of existing verifiers to provide the necessary reward signal for RL-based training.
- Demonstrating that RL with verifier feedback can train effective specification generators without human preference judgments or chain-of-thought supervision. The verifier serves as an automated oracle that replaces RLHF
- A novel *subset reward* that measures whether generated specifications are logically stronger than ground truth by checking precondition relaxation and postcondition strengthening to address reward hacking where models produce trivial/weak but verifiable specifications
- An automated data curation pipeline that converts Python code to verified Dafny programs using frontier LLMs, eliminating per-example human annotation
- DafnyComp, a benchmark for specification reasoning that consists of synthetic Dafny programs with auto-formalized ground truth specifications

**Additional Comments:**

N/A

**Audience:**

Yes

**Audience Explanation:**

This work is definitely interesting to TMLR's audience. It addresses the challenge of obtaining reliable training signals for reasoning tasks by grounding LLMs in formal verification systems. The resulting training pipeline, especially their proposed subset reward mechanism, offers a principled approach to measuring specification quality that could inform reward design in other domains where outputs have logical structure.

**Broader Impact Concerns:**

No concerns

**Claims And Evidence:**

Yes

**Claims Explanation:**

The claim that verifier-guided RL produces effective specification generators is overall well-supported. As the authors acknowledge, eligible Dafny code is scarce, with only 1.2k samples remaining after filtering public repositories. As a result, frontier models have likely had minimal exposure to Dafny, so fine-tuning a base model on domain-specific data predictably outperforms non-fine-tuned LLMs. The more meaningful evidence comes from the SFT-to-RL comparisons, which demonstrate consistent improvements (for pass@1 and pass@128 with entropy regularization) from the novel verifier-guided training approach.
In their ablations the authors show that models trained only with verification reward achieve high verification rates but produce trivial/weak specifications that game the metric, while the subset reward maintains specification quality. This clearly demonstrates the necessity of their reward design. The ablations also establish that entropy regularization is key for boosting exploration, which drives pass@128 performance.

**Requested Changes:**

This is not critical to my recommendation, but I think the phrasing as "reducing human priors" is somewhat vague and distracts from the actual point. While per-example annotation is eliminated and human preference judgments are replaced by verifier feedback, substantial human knowledge remains embedded in the base model pretraining and the Claude-generated training data. I believe the contribution is better characterized as replacing human feedback with formal verification rather than eliminating human priors.

---

> ### Author Response · Authors · 2026-04-09
> **Response to Reviewer v3Qt**
>
> *Note: We have uploaded a revised manuscript based on your feedback. For your convenience, all new additions and major revisions in the updated PDF have been highlighted in **purple**.*
>
> We sincerely thank the reviewer for the thoughtful evaluation and constructive suggestions, which have helped us significantly strengthen the paper. We agree that there remain indirect human priors from pretrained models, prompt templates, and the data synthesis process, and that our phrase `reducing human priors` is too vague if read literally. Accordingly, we have revised our framing to make a more precise claim: our method replaces human feedback with formal verifier feedback, thereby reducing reliance on direct human annotations. We have updated this terminology throughout the revised manuscript, specifically in the title, abstract, introduction, and conclusion, to ensure our framing accurately and rigorously reflects our contributions.

---

### Review · Reviewer_oae9 · 2026-03-08

**Summary Of Contributions:**

### Summary of contributions
This paper introduces Re:Form, a reinforcement learning framework designed to train LLMs to generate formal specifications in Dafny so that both the reasoning process and the generated code can be formally verified.
The authors present:
- An automated data curation pipeline to construct a Dafny training dataset,
- A new benchmark, DafnyComp, with auto-formalized specifications,
- A Dafny-specific training pipeline that introduces a novel reward mechanism and a new metric, Spec Superiority Rate (SSR), designed to prevent trivial verifications and encourage exploration and generalization.

Results show that relatively small models trained with this framework can outperform larger frontier models on the newly introduced benchmark.

### Strenghts
1. The problem is well motivated. As spec-driven development and LLM-generated code become more widespread, verification becomes increasingly important. Tests alone are inherently incomplete. This work explores an alternative direction in which LLM-generated code is formally verifiable.
2. The introduction of Spec Superiority Rate (SSR) and the subset-reward mechanism are relevant contributions to verifier-based RL training. They address reward hacking and promote stronger specifications. DafnyComp is also a useful addition to the ecosystem and is made open-source.
3. The experimental framework is carefully designed. The experiments analyze both in-domain and out-of-domain performance. The ablation study provides insight into the effect of different reward configurations and regularization strategies, even though some aspects remain unclear (see below).
4. The paper includes an analysis of novel specification discovery under RL, which strengthens the empirical contribution and provides additional evidence that exploration is occurring beyond simple memorization.

### Weaknesses
1. As someone unfamiliar with Dafny and formal specifications, I am concerned that most ground-truth specifications in the data curation pipeline are generated automatically using Claude. While the filtering and verification process is described in detail, these ground truths are central to SSR and the benchmark itself. The paper would be strengthened by including expert validation.
2. Chain-of-thought (CoT) reasoning is dismissed in multiple parts of the paper, yet there is no direct empirical demonstration that CoT would not help in this setting. The authors acknowledge this partially in the conclusion. Additionally, it is unclear whether frontier baselines are evaluated under their strongest configurations. Since frontier models often benefit significantly from CoT or multi-step prompting, removing it may disadvantage those models.
3. Although improvements over baselines are reported, the absolute pass@1 performance remains relatively low. It is not clear what these numbers imply for practical formal verification in real-world scenarios.
4. No failure analysis.
5. As someone familiar with current coding benchmarks such as SWE-Bench or FEA-Bench, it is difficult to see how this work integrates into that landscape. There appears to be a gap between formal specification generation and contemporary repository-level or bug-fixing benchmarks. A clearer bridge to those settings would strengthen the work's positioning.
6. In the ablation study, entropy appears to contribute to instability and collapse in the training curve. Its overall contribution is therefore ambiguous, and the trade-off between diversity and stability could be clarified further.

**Additional Comments:**

N/A

**Audience:**

Yes

**Audience Explanation:**

The paper addresses a timely and relevant problem for the TMLR community. Formal verification represents a principled direction as AI-generated code becomes more prevalent. The release of a new benchmark further contributes to the research ecosystem and may facilitate future work in this space.

**Claims And Evidence:**

Yes

**Claims Explanation:**

The main claims of the paper are supported by clear and generally convincing evidence within the presented experimental setting. The framework is described in sufficient detail, and the introduced methodological components are well motivated. The authors provide both qualitative and quantitative analyses that help substantiate their contributions.

**Requested Changes:**

### Critical Changes
- Address Weakness 1: Strengthen ground-truth validation, either by introducing expert review (preferably on at least a subset of the data) or by expanding the analysis of the generated specifications to better justify their quality and completeness.
- Address Weakness 3 and 5: Improve the positioning of this work with respect to real-world use cases. In particular, clarify how commonly used benchmarks (e.g., SWE-Bench and similar settings) could benefit from this approach, or explicitly argue that the two paradigms are orthogonal.

### Minor Changes:
- Address Weakness 2: Tone down the claims regarding CoT or provide empirical justification.
- Address Weakness 4: Given the novelty of the topic and the relatively low absolute performance numbers, a more detailed failure analysis is needed. For example: what kinds of specifications fail, and why?
- Address Weakness 6: Clarify the contribution and trade-offs of the entropy term.

---

> ### Author Response · Authors · 2026-04-09
> **Response to Reviewer oae9**
>
> *Note: We have uploaded a revised manuscript based on your feedback. For your convenience, all new additions and major revisions in the updated PDF have been highlighted in **purple**.*
>
> We sincerely thank the reviewer for the constructive feedback, which has significantly strengthened our paper. Below, we address your comments point by point.
>
> **1. Quality of ground-truth specifications:** We agree that the quality of the auto-generated reference specifications needs to be justified more clearly. To make this more concrete, we performed a corpus-level structural analysis over the full python2dafny-18k ground-truth set (17,521 programs). As shown in Figure: https://postimg.cc/vg8PTWJs, across the full corpus, the references contain 72,141 `requires`, 136,495 `ensures`, 49,855 loop invariants, 15,957 `modifies`, 5,886 `reads`, and 11,560 `assert` statements. At the example level, 84.6% of programs contain both preconditions and postconditions, 52.7% contain quantified or old-state reasoning (`forall` / `old(...)`), and 54.4% include frame or heap-sensitive specifications (`modifies`, `reads`, or `fresh`). At the structural level, 87.6% of methods and 90.4% of constructors receive at least one explicit contract, and 98.0% of loops receive at least one invariant, with an average of 2.69 invariants per loop. These statistics do not replace expert validation, but they do show that the references are systematically rich, non-trivial, and closely aligned with the program structure rather than being sparse verifier-passing placeholders.
>
> **2. Connection to real-world settings.** We agree that the paper should better explain how this line of work could connect to practical software settings. In the revision, we have added a discussion (Section 4) and qualitative examples (Appendix B.5) illustrating how formal specification generation can support real-world Python code generation and verification workflows, despite our focus on a controlled Dafny setting rather than full repository-level tasks. To make this point more concrete, we ran our own inference pipeline with the trained ReForm model on 6 small translated Python-style bug patterns covering hidden preconditions and semantic logic errors. In all 6 cases, the model produced non-trivial contracts or invariants rather than empty verifier-passing placeholders; in 3 cases the generated specification verified successfully, and in 3 cases it caused Dafny to reject the buggy implementation. Among the clearest examples, the model generated `requires |xs| > 0` and `ensures y == xs[0]` for a sequence-head example, making the hidden non-empty-input assumption explicit; it generated `requires x != 0` and `ensures y == 100 / x` for a division example, surfacing a hidden arithmetic precondition; and for buggy absolute-value and max-style implementations it generated semantic postconditions such as `ensures y >= 0` and `ensures z >= y`, after which Dafny rejected the code because those intended properties could not be proved. These small runs are not a substitute for a repository-level benchmark, but they do support the practical claim that generated formal specifications can act as a debugging layer by surfacing hidden assumptions and semantic inconsistencies in generated or human-written Python code. We view this as complementary rather than directly comparable to repository-level benchmarks such as SWE-Bench: those benchmarks test end-to-end bug fixing in large codebases, whereas our setting isolates whether a model can infer formal contracts that expose hidden assumptions or semantic mismatches.
>
> **3. CoT discussion:** We agree that our wording around CoT should be softened. Our current evidence only shows that visible CoT in current commercial LLMs does not solve this problem reliably, and that our own gains do not rely on human-labeled CoT supervision. This should not be interpreted as a general claim that CoT is unnecessary or ineffective in all settings, as we mentioned in Section 4. We will revise the paper accordingly.
>
> **4. Failure analysis:** We agree that an explicit failure analysis strengthens the paper, and we have added a dedicated section (Appendix B.6) to the revised manuscript. As summarized in Figure: https://postimg.cc/vg8PTWJs, the dominant failures are weak but verifier-passing specifications (52.2%) and unverifiable specifications (44.3%), while trivial specifications are rare (3.5%). This new appendix section provides a finer-grained taxonomy of these errors alongside representative examples.
>
> **5. Entropy regularization:** The current evidence in Figure 1 suggests that entropy regularization has both value and cost: it improves exploration and pass@k, but it also destabilizes training and can encourage useless trailing-token emission. We will revise the text so that this trade-off is stated clearly and not presented as an unqualified advantage.

---

### Review · Reviewer_CT5a · 2026-03-24

**Summary Of Contributions:**

The paper proposes a framework for formal specification generation using LLMs trained with minimal human priors and reinforcement learning (RL), focusing on the Dafny verification language. Its central idea is to reduce reliance on human supervision by eliminating human-annotated data, natural language chain-of-thought, and human-based reward signals, instead leveraging pretrained models with light supervised fine-tuning, automatically generated data, and formal verification signals as supervision—thereby shifting from human-guided reasoning to self-improving agents grounded in formal systems. To support this, the paper introduces a fully automated end-to-end pipeline that converts Python programs into Dafny with auto-generated specifications, uses an iterative verification-and-repair loop, and produces around 20k verified programs without per-example human annotation, highlighting formal verification as a scalable supervision signal. It further develops a reinforcement learning framework with verifier-based rewards, combining syntax and verification rewards with a novel subset (spec superiority) reward that encourages stronger specifications, optimized via GRPO, thus moving beyond merely passing verification to improving specification quality via logical dominance. The paper also introduces DafnyComp, a synthetic benchmark designed to evaluate compositional formal reasoning, including in-domain, out-of-domain, and multi-function generalization settings, addressing limitations of prior benchmarks. Empirically, the results show that relatively small models (up to 14B parameters) can generate verifiable formal code and outperform proprietary LLMs on this task, with RL improving over supervised fine-tuning in both verification rate and spec superiority rate, enabling discovery of novel specifications and stronger out-of-domain generalization (e.g., ~14% vs 8.3% baseline without RL). Overall, the work provides evidence that RL-driven exploration yields genuine reasoning improvements rather than mere memorization, and advances the broader insight that formal languages offer sound, automatic feedback signals that can enable scalable, closed-loop learning for autonomous reasoning systems with reduced dependence on human priors.

Strengths:
- The goal of improving scalability of learning reasoning verifiers by reducing the need for human-annotated data is very well-motivated.
- The use of formal verification avoids potential noise in human data.
- Experiments show the advantage of using RL over supervised fine-tuning.
- The evaluation metric “spec superiority rate” which measures the fraction of generated specs that are at least as good as the ground truth seems appropriate and useful.

Weaknesses:
- The use of formal language based approach which is highly structured makes it unclear if the techniques will work for natural language reasoning or more open-ended planning tasks.
- The evaluation is primarily on synthetic data, so it is not clear if there are further obstacles if the approach were to be applied to real-world data.
- The evaluation of the approach still needs ground-truth data.
- The absolute performance is only ~14% on OOD data. Of course this is a very challenging setting and the approach beats the baseline.

**Additional Comments:**

Is it appropriate to call the approach a form of self-supervised learning for formal reasoning verification? It might be worth adding a discussion how the approach differs from self-supervised approaches in the literature for other non-reasoning tasks.

**Audience:**

Yes

**Audience Explanation:**

Yes, at least some individuals in TMLR’s audience would likely be interested. The paper sits at an intersection that TMLR readers care about: LLM reasoning, reinforcement learning, code generation, and formal verification.

**Broader Impact Concerns:**

I think it is important to caution that reducing human priors may remove valuable expert judgment and preferences and one must consider the ethical implications of doing this for other possible application areas.

**Claims And Evidence:**

Yes

**Claims Explanation:**

The paper gives clear empirical evidence for its core experimental claims. It describes the pipeline in detail, defines the training stages and reward schemes, and reports results on in-domain data and a new out-of-domain benchmark. It also includes ablations showing that verification-only rewards can be gamed by producing weak specifications, while the proposed subset-based reward improves specification quality, which directly supports one of the paper’s central methodological claims. The paper further gives qualitative examples of “novel” specifications found after RL and reports scaling trends across model sizes, which makes the evidence fairly convincing for the claim that RL helps beyond SFT in this formal-language environment.

The conceptual claims are less clear to me. A central claim is to reduce “human priors”. While I agree with scalability and efficiency in using the formal-language based approach, there are still human priors in the form of prompt templates, reward design, and frontier models for data generation. Perhaps the framing should instead be in terms of reducing tedious human annotation of chain-of-thought instead? Also, frontier LLMs are used for generating formal specifications and these LLMs likely use CoT to generate these specs, so it is not clear if CoT is actually completely avoided in this approach.

**Requested Changes:**

- The claim around reducing human priors and implications for reasoning should be carefully qualified, as the system still relies heavily on pretrained models, prompt design, and synthetic data generation.
- Is it possible to evaluate on real-world datasets? If not, add a discussion why synthetic data is representative and what are possible limitations.
- The novel contributions relative to prior work on program synthesis with execution feedback and proof assistants + RL / search should be clarified.
- Characterize when the model fails, whether it is due to incorrect specs vs unverifiable vs trivial specs.
- How sensitive are results to initial seed data quality and the choice of frontier model?

---

> ### Author Response · Authors · 2026-04-09
> **Response to Reviewer CT5a**
>
> *Note: We have uploaded a revised manuscript based on your feedback. For your convenience, all new additions and major revisions in the updated PDF have been highlighted in **purple**.*
>
> We sincerely thank the reviewer for the thoughtful evaluation feedback. Below, we address your questions point by point.
>
> **1. Scope of the claim:** Our work focuses on formal language because it is highly structured and strictly verifiable. We agree that our current results should not be interpreted as evidence that the same approach will automatically transfer to natural-language reasoning or more open-ended planning tasks. In the revision, we will narrow this claim and make it explicit that this paper is a pilot study on formal specification generation in Dafny.
>
> **2. Real-world data:** We translate Python programs into Dafny mainly because publicly available real-world Dafny data is very limited. At the same time, our evaluation is not purely synthetic: in addition to the in-domain and DafnyComp settings, we also evaluate on the real-world benchmark DafnyBench. As shown in Figure 8 in the paper, our model achieves higher verification rate and SSR than strong baseline LLMs on this benchmark as well. We will strengthen this point in the revision and better explain the role of synthetic data as a controlled testbed rather than a claim of full real-world coverage.
>
> **3. Evaluation using ground truth:** Two of our evaluation metrics, validity rate and verification rate, can be computed directly from the Dafny verifier and do not require ground-truth specifications. However, once the question becomes whether a verified specification is also non-trivial and behaviorally strong, the verifier alone does not provide an absolute signal. In this setting, we only have a comparative notion of quality, namely whether one specification is weaker than, comparable to, or stronger than another. This is why we use ground truth for SSR: not because it is perfect, but because it provides a practical reference point for evaluating specification strength beyond mere verifiability.
>
> **4. Novelty relative to prior work:** We agree that the high-level SFT + RL training recipe is not by itself novel. Our main novelty lies in adapting this framework to formal specification generation with a reward design that is tailored to formal verification. In particular, the subset reward is designed to distinguish semantically meaningful specifications from weak but verifier-passing ones, which is a failure mode not addressed by verification rate alone. We will clarify this distinction more explicitly in the related-work section.
>
> **5. Failure analysis:** We agree that the paper would benefit from a more explicit characterization of failure modes. We therefore conducted a qualitative review of 115 semantically analyzable failures from ReForm-3B-RL on Python2Dafny. As summarized in Figure: https://postimg.cc/vg8PTWJs, 52.2% are weak but verifier-passing specifications, 44.3% are unverifiable specifications, and only 3.5% are trivial specifications. The dominant unverifiable errors are unproved postconditions and non-inductive loop invariants, while the dominant weak-specification errors are omitted semantic assertions or missing subset obligations relative to the reference. We have added a detailed failure analysis in Appendix B.6 of the revised manuscript, which includes finer-grained statistics and representative examples. The main takeaway is that current failures are driven predominantly by semantic weakness and proof misalignment, rather than by degenerate reward hacking.
>
> **6. Sensitivity to seed quality:** We thank the reviewer for raising this point. We have not yet performed a controlled sensitivity study across different frontier models or seed-data generators. All reported results use the same Qwen2.5-Coder-based pipeline to produce the initial seed data, so our current evidence only supports stability with respect to RL sampling randomness, not robustness to changing the seed-data generator itself. We will state this limitation explicitly in the revision.

---

### Decision · Action_Editor_c9o6 · 2026-04-30

**Recommendation:** Accept as is

**Audience:**

Yes

**Audience Explanation:**

This work presents contributions in many interrelated fields such as language models, code generation, and formal problem verification, that would be interesting to many individuals in the TMLR's audience.

**Claims And Evidence:**

Yes

**Claims Explanation:**

After the review, all claims were either supported by convincing evidence, or clarified or moved to limitations.